# Spatio-temporal relays control layer identity of direction-selective neuron subtypes in *Drosophila*

Holger Apitz [1] & Iris Salecker [1]

Visual motion detection in sighted animals is essential to guide behavioral actions ensuring their survival. In *Drosophila*, motion direction is first detected by T4/T5 neurons. Their axons innervate one of the four lobula plate layers. How T4/T5 neurons with layer-specific representation of motion-direction preferences are specified during development is unknown. We show that diffusible Wingless (Wg) between adjacent neuroepithelia induces its own expression to form secondary signaling centers. These activate Decapentaplegic (Dpp) signaling in adjacent lateral tertiary neuroepithelial domains dedicated to producing layer 3/4-specific T4/T5 neurons. T4/T5 neurons derived from the core domain devoid of Dpp signaling adopt the default layer 1/2 fate. Dpp signaling induces the expression of the T-box transcription factor Optomotor-blind (Omb), serving as a relay to postmitotic neurons. Omb-mediated repression of Dachshund transforms layer 1/2- into layer 3/4-specific neurons. Hence, spatio-temporal relay mechanisms, bridging the distances between neuroepithelial domains and their postmitotic progeny, implement T4/T5 neuron-subtype identity.

[1] The Francis Crick Institute, Visual Circuit Assembly Laboratory, 1 Midland Road, London NW1 1AT, UK. Correspondence and requests for materials should be addressed to I.S. (email: iris.salecker@crick.ac.uk)

Visual signals received by the retina are generally not stationary because objects in the environment and/or the bodies of animals move. To detect motion, visual circuits perform complex spatio-temporal comparisons that convert luminance changes collected by photoreceptors into signals containing information about direction or speed. Despite the seemingly divergent anatomy of vertebrate and insect visual systems, they display remarkable parallels in the computations underlying motion vision and the neuronal elements performing them[1,2]. In most sighted animals, this involves neurons that respond to motion signals in specific directions. Direction-selectivity emerges from differences in the connectivity of their dendrites[2]. Motion-direction preferences by their axons are represented by layer-specific innervation[3–8]. Thus, anatomical characteristics such as layer-specificity seem to be intricately linked with motion-directionality. However, how these are implemented during circuit development is poorly understood.

The *Drosophila* visual system has emerged as a powerful model for elucidating the neural circuits and computations underlying motion detection. Photoreceptors (R-cells) in the retina extend axons into the optic lobe consisting of the lamina, medulla, lobula plate, and lobula (Fig. 1a). Neuronal projections in these ganglia are organized into retinotopically arranged columnar units. The medulla, lobula plate, and lobula are additionally subdivided into synaptic layers. They are innervated by more than a 100 neuronal subtypes that extract different visual features in parallel pathways[9]. T4 and T5 lobula plate neurons are the first direction-selective circuit elements[6,10]. Each optic lobe hemisphere contains ~5300 T4/T5 neurons[11]. T4 dendrites arborize within medulla layer 10, and T5 dendrites in lobula layer Lo1. Their axons project to one of the four lobula plate layers, thereby defining four different neuron subtypes each[12] (Fig. 1a). Axons segregate according to their motion-direction preferences. Thus, front-to-back, back-to-front, upward, and downward cardinal motion directions are represented in lobula plate layers 1–4[5,6]. T4 neurons are part of the ON motion detection pathway reporting brightness increments, while T5 neurons are part of the OFF pathway reporting brightness decrements[6]. Distinct neuron sets in the lamina and medulla relay ON and OFF information to T4 and T5 neurons[2,13]. Direction-selectivity emerges within T4/T5 dendrites and involves the non-linear integration of input from these upstream neurons for enhancement in the preferred direction and suppression in the null-direction[10,14–16]. Dendritic arbors of the four T4 neuron subtypes have characteristic orientations, that correlate with the direction preferences of lobula plate layers innervated by their axons[17,18]. Thus, direction-selectivity involves the establishment of neuron subtypes, each with distinct spatial connectivities. Here, we address when and how T4 and T5 neuron subtypes with different layer identities are specified during development.

Optic lobe neurons originate from two horseshoe-shaped neuroepithelia, called the outer and inner proliferation centers (OPC and IPC; Fig. 1b)[19,20]. These are derived from the embryonic optic lobe placode[21] and expand by symmetric cell divisions during early larval development[22,23]. At the late 2nd instar larval stage, neuroepithelial (NE) cells from the medial OPC edge begin to transform into medulla neural stem cells, called neuroblasts (Nbs)[20]. These undergo asymmetric divisions to self-renew and give rise to ganglion mother cells (GMCs), which divide to generate two neurons or glia[22,24]. Apposing the OPC, two dorsal and ventral NE domains, called the glial precursor cell (GPC) areas, produce neuron subtypes associated with all ganglia[25,26]. At the mid 3rd instar larval stage, the lateral OPC begins to generate lamina neurons[20].

The IPC generates lobula and lobula plate neurons, including T4/T5 neurons from the early 3rd instar larval stage onward[20]. Our recent studies showed that NE cells in one domain, the proximal (p-)IPC, convert into progenitors in an epithelial-mesenchymal transition (EMT)-like process[23,27]. Progenitors migrate to a second proliferative zone, the distal (d-)IPC, where they mature into Nbs. These transition through two competence windows to first produce C&T neurons, corresponding to C2 and C3 ascending neurons connecting the medulla and lamina, as well as T2/T2a and T3 neurons connecting the medulla and lobula[12], and then T4/T5 lobula plate neurons (Fig. 1a, b). Cross-regulatory interactions between Dichaete (D) and Tailless (Tll) control the switch in Nb competence defined by the sequential expression of the proneural bHLH transcription factors Asense (Ase) and Atonal (Ato). The latter is co-expressed with the retinal determination protein Dachshund (Dac)[23]. The molecular mechanisms that control layer-specific T4/T5 neuron subtype identities within this sequence of developmental events occurring at different locations have remained elusive.

T4/T5 neuron diversity resulting in differential layer-specificity could be achieved by postmitotic combinatorial transcription factor codes upstream of distinct guidance molecules. Although not mutually exclusive, layer-specificity of T4/T5 neurons could also be determined by temporal differences in the expression of common postmitotic determinants, similar to the birth-order dependent R-cell growth cone segregation strategy described in the medulla[28,29]. Here, we provide evidence for another mechanism, whereby layer-specific T4/T5 neuron subtype identity is determined early in the p-IPC neuroepithelium. Their specification depends on two relay mechanisms involving Wnt and Bone morphogenetic protein (Bmp) signaling and transcription factor interactions. These establish and translate the spatial patterning of NE cells into postmitotic neuronal subtype identities to bridge distances inherent to this particular neurogenesis mode.

## Results

**Layer 3/4 innervating T4/T5 neurons depend on Wg secretion.** Wnt family members are evolutionary conserved signaling proteins that orchestrate tissue patterning and growth during development. Recent findings showed that flies expressing membrane-tethered instead of normally secreted Wingless (Wg) were viable and had normal bodies and well-patterned, albeit slightly smaller wings[30]. This indicated that long-range spreading of Wg is not essential for the development of many tissues. As the general fitness of these flies was reduced, Wg release could possibly be required in tissues other than imaginal discs. In the visual system, Wg is expressed and required in the GPC areas for neuron specification[25,26,31,32]. To explore whether Wg plays a role in the IPC and the spreading of this signaling molecule is essential, we examined the brains of adult homozygous flies, engineered to solely express Wg fused to the type-2 transmembrane protein Neurotactin[30] (*wg{KO;NRT-wg}*; Fig. 1c). We uncovered a specific and highly penetrant phenotype in the lobula plate (Fig. 1d–g): unlike in controls ($n = 13/13$), either one ($n = 18/30$) or two ($n = 12/30$) of the four lobula plate layers were missing. Moreover, T4/T5 neuron numbers were reduced by approximately 25% in three-layered and 50% in two-layered samples, respectively (Fig. 1h). To determine which layers were affected, optic lobes were immunolabeled with the cell surface molecule Connectin, a specific marker of lobula plate layers 3/4. In contrast to controls ($n = 26/26$), Connectin was expressed either in one ($n = 16/30$) or none ($n = 14/30$) of the lobula plate layers in *wg{KO;NRT-wg}* flies (Fig. 1i–k). Thus, T4/T5 neurons innervating layers 3/4 were preferentially affected in flies solely expressing membrane-tethered Wg.

To elucidate the underlying causes, we examined the expression of wild-type Wg in the 3rd instar larval optic lobe. Consistent with previous reports[26], Wg protein was detected in the dorsal and ventral GPC areas adjacent to the OPC (Fig. 1l, n, o). Apposing the ventral p-IPC shank, Wg was expressed in surface (s-)IPC NE cells, which generate two lobula neuron clusters[23]. Additionally, Wg was expressed in a small Nb clone adjacent to the dorsal p-IPC shank (Fig. 1m–o). Relying on the perdurance of reporter gene expression by two wg-Gal4 drivers to mark the progeny of these domains

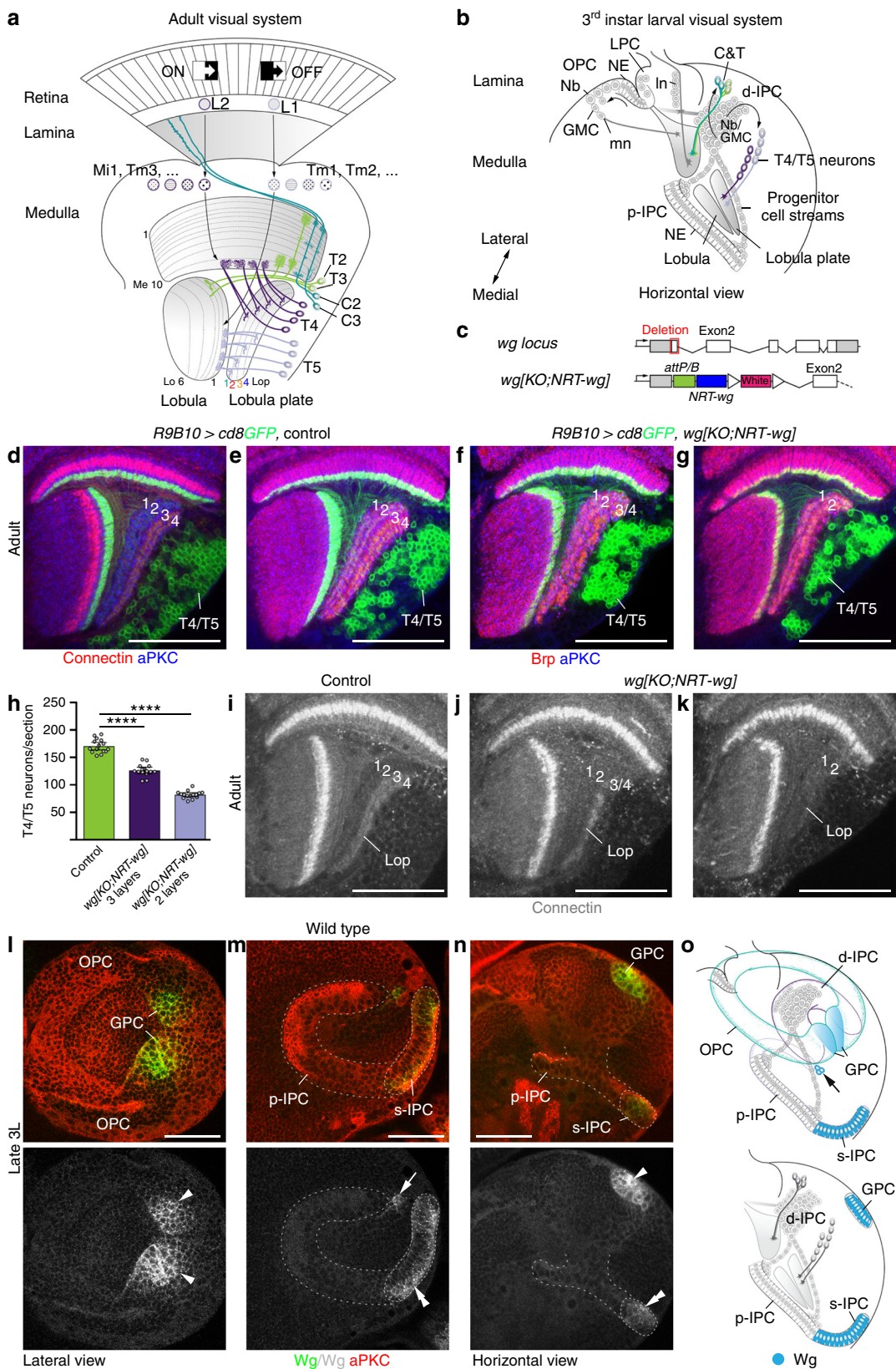

(Supplementary Fig. 1a–d), we observed that *wg* is not produced in T4/T5 neurons or their progenitors in the d-IPC and p-IPC. This suggests a non-autonomous requirement of Wg for the development or survival of T4/T5 neurons innervating layers 3/4.

**Wg secretion from the GPC areas induces *wg* in secondary domains.** We next asked how the prevention of Wg release might cause the lobula plate-specific defects. In the GPC areas, Wg blocks the expression of the transcriptional regulator Homothorax (Hth), a temporal series factor in the OPC[32]. Hth expression was correctly suppressed in *wg{KO;NRT-wg}* flies, suggesting that NRT-Wg functions as wild-type Wg in this context (Supplementary Fig. 2a, b). However in the IPC (Fig. 2a, b; Supplementary Fig. 2c–f), NRT-Wg expression was absent in the s-IPC adjacent to the ventral p-IPC and the Nb clone adjacent to the dorsal p-IPC in approximately half of the samples (n = 22/41). In the remaining samples, residual NRT-Wg expression was found either in the s-IPC (n = 9/41), the Nb clone (n = 8/41), or both (n = 2/41). The s-IPC and its progeny developed normally, because Hth was expressed and two lobula neuron clusters were formed (Supplementary Fig. 2g–j). Wg protein and reporter gene expression were detected in the GPC areas before the s-IPC in late 2nd instar larvae (Fig. 2c, d). Furthermore, *wg* expression in the s-IPC of *wg{KO;NRT-wg}* mid 3rd instar larvae was absent when *wg* normally is detected (Fig. 2e, f). Thus, GPC-derived Wg could induce Wg expression in the s-IPC and Nb clone.

Failure to induce NRT-Wg expression could conceivably not be due to lack of Wg release but instead to sub-optimal signaling activity of the NRT-Wg protein. We therefore asked whether boosting NRT-Wg expression could overcome observed defects. This was tested by combining a *UAS-NRT-wg* transgene, known to be highly active[33] with the knock-in null allele *wg{KO;Gal4}*, serving also as a driver that restricts activation to endogenous expression domains[30]. *UAS-FLP* and *wg{KO;FRT wg+ FRT NRT-wg}*[30] ensured that the endogenous locus only produced NRT-Wg following the induction of recombination events. A *tub-Gal80ts* transgene controlled the timing of expression (for full genotypes, see Supplementary Table 1). Thus, at a restrictive temperature, Wg expression is normal and no NRT-Wg is produced. After a shift to a permissive temperature, *UAS-NRT-wg* is activated and the *wg* allele switches to expressing NRT-Wg. We found that following a temperature shift at the mid 3rd instar larval stage, NRT-Wg was expressed in the GPC areas and the s-IPC (Fig. 2g; n = 16/16). However, following an early temperature shift at the 1st instar larval stage, no expression could be detected in the s-IPC in the majority of samples (76%; n = 26/34) (Fig. 2h). We suggest that this partial phenotype is due to imperfect allele switching, i.e., residual wild-type Wg is produced by the GPC areas, and that Wg release from the GPC areas is required

between the 1st and mid 3rd instar larval stage for inducing Wg expression in the s-IPC. Thus, impaired Wg release and not reduced signaling activity in *wg{KO;NRT-wg}* flies accounts for the loss of NRT-Wg expression in secondary domains.

Next, we performed converse allele switching experiments using *wg{KO;FRT NRT-wg FRT wg+}* to express wild-type Wg in a *wg{KO;NRT-wg}* background[30]. We used *R46E01-Gal4* (Supplementary Fig. 2k, l) and *UAS-FLP* to induce recombination specifically in the GPC areas. Expression of wild-type *wg* in the GPC areas rescued the s-IPC-specific loss of NRT-Wg (Fig. 2i, j; n = 16/23), demonstrating that Wg release specifically from the GPC areas is essential.

Consistent with Wg signaling-dependent induction of *wg*, the target genes *frizzled 3* (*fz3*) and *notum* were expressed in the s-IPC (Fig. 2k, l). Moreover, Wg expression in the s-IPC was abolished following the simultaneous knockdown of the Wg receptors *frizzled* (*fz*) and *frizzled 2* (*fz2*) (Fig. 2m, n) in the IPC by combining the *fas3-Gal4* driver[23] (cf. Supplementary Fig. 7a) with UAS-RNA interference (RNAi) transgenes. Hence, Wg released from the GPC areas is required to induce *wg* in the s-IPC and Nb clone (Fig. 2o).

**wg is required to induce *dpp* in adjacent p-IPC subdomains.** How can *wg* in these secondary domains control T4/T5 neurogenesis in the p-IPC/d-IPC? The *Drosophila* Bmp family member Decapentaplegic (Dpp) is a known target of *wg* in the OPC and is expressed adjacent to the GPC areas in dorsal and ventral OPC subdomains[26] (Fig. 3a). Similarly in the IPC, *wg* expression in the s-IPC and Nb clone was detected adjacent to *dpp*-positive ventral and dorsal p-IPC subdomains (Fig. 3b–f). *dpp* reporter gene expression persisted in the two progenitor streams arising from these subdomains[23] (Fig. 3c, d). Consistent with stepwise inductive events, *dpp* in the p-IPC did not precede *wg* expression in the s-IPC in late 2nd instar larvae (Fig. 3e). Furthermore, *fz3* and *notum* (Fig. 2k, l) were expressed similarly to *dpp* in p-IPC subdomains and progenitor streams. In *wg{KO;NRT-wg}* flies, *dpp-lacZ* in the p-IPC was either absent (Fig. 3g, h; n = 16/32) or showed only residual labeling in one progenitor stream (Supplementary Fig. 3a; n = 16/32), in line with the penetrance and expressivity of phenotypes observed in adults. By contrast, OPC expression was unaffected, suggesting that releasable Wg is not required in this region (Fig. 3g, h). Constitutively active Wg signaling induced by IPC-specific expression of Armadillo[S10] resulted in ectopic *dpp-lacZ* labeling (Fig. 3i; Supplementary Fig. 3b), confirming that *dpp* is a Wg target in the p-IPC. IPC-specific *wg* knockdown abolished *dpp-lacZ* labeling, corroborating that *wg* is required in the s-IPC and Nb clone, and not the GPC areas for induction (Fig. 3j). Since Dpp signaling mediates EMT of migratory progenitors in the Dpp-expression domains[23], cell

**Fig. 1** Wg release is essential for the formation of lobula plate layers 3/4. **a** Schematic of the adult *Drosophila* visual system. Neurons in the lamina (L1/L2) and medulla (e.g., Mi1,4,9, Tm1–4,9) relay ON/OFF motion cues to T4 and T5 neuron dendrites in medulla layer (Me) 10 and lobula (Lo) layer 1. T4/T5 axons innervate lobula plate (Lop) layers 1–4. C&T neurons include C2/C3 and T2/T3 subtypes. **b** Schematic of the 3rd instar larval optic lobe. The OPC generates lamina (ln) and medulla (mn) neurons. p-IPC NE cells give rise to migratory progenitors that mature into d-IPC Nbs. These produce C&T and T4/T5 neurons. GMC ganglion mother cells, LPC lamina precursor cells. **c** Structure of wild-type *wg* and engineered *wg* loci (*wg{KO;NRT-wg}*). Open triangles indicate *loxP* sites. **d** R9B10-Gal4 UAS-cd8GFP (green) labels T4/T5 neurons. Connectin (red) marks Lop layers 3/4. **d–g** Neuropils were stained with nc82 (red) and aPKC (blue). Compared to controls (**e**), in *wg{KO;NRT-wg}* flies, one (**f**) or two (**g**) lobula plate layers were absent. **h** The decrease of layers correlates with T4/T5 neuron numbers. The scatter plot with bars shows data points and means with ±95% confidence interval error bars (n = 15; three optical sections from five samples per genotype). Unpaired, two-tailed Student's *t*-test not assuming equal variance: P = 4.72 × 10⁻¹¹ and P = 3.23 × 10⁻¹⁷. ****P < 0.0001. Unlike in controls (**i**), Connectin was found in one (**j**) or none (**k**) of the Lop layers in *wg{KO;NRT-wg}* flies. Similar to nc82 (**f**), Connectin labeling showed gaps in the third lobula plate layer (**j**), potentially consisting of both layer 3 and 4 neurons. **l–n** In wild-type 3rd instar larvae (3L), the GPC areas (arrowheads), surface (s-)IPC (dashed line, double arrowheads), and a Nb clone (arrow) adjacent to the dorsal p-IPC subdomain (dashed line) express Wg (green). **o** Schematics of larval Wg expression (blue) in 3D and a horizontal section. Arrow indicates Nb lineage. For genotypes and sample numbers, see Supplementary Table 1. Scale bars, 50 μm

streams from these areas were affected in wg{KO;NRT-wg} flies and following IPC-specific knockdown of wg and the Dpp type I receptor thickveins (tkv) (Fig. 3h, j, k). Although the overall d-IPC morphology was altered (Supplementary Fig. 3c–f), progenitors and Nbs generated from the remaining p-IPC showed wild-type marker expression (Supplementary Fig. 3g–n). In adults, similar to wg{KO;NRT-wg} flies, lobula plate layers 3/4 were absent following IPC-specific knockdown of fz, fz2, or tkv (Fig. 3l–n).

Hence, wg from the s-IPC and Nb clone regulates dpp expression in adjacent p-IPC subdomains and EMT of progenitors that mature into Nbs producing T4/T5 neurons for layers 3/4 (Fig. 3o).

**Layer 3/4 T4/T5 neurons arise from Dpp-positive subdomains.** To provide evidence that T4/T5 neurons innervating layers 3/4 specifically originated from the Dpp-positive subdomains in the

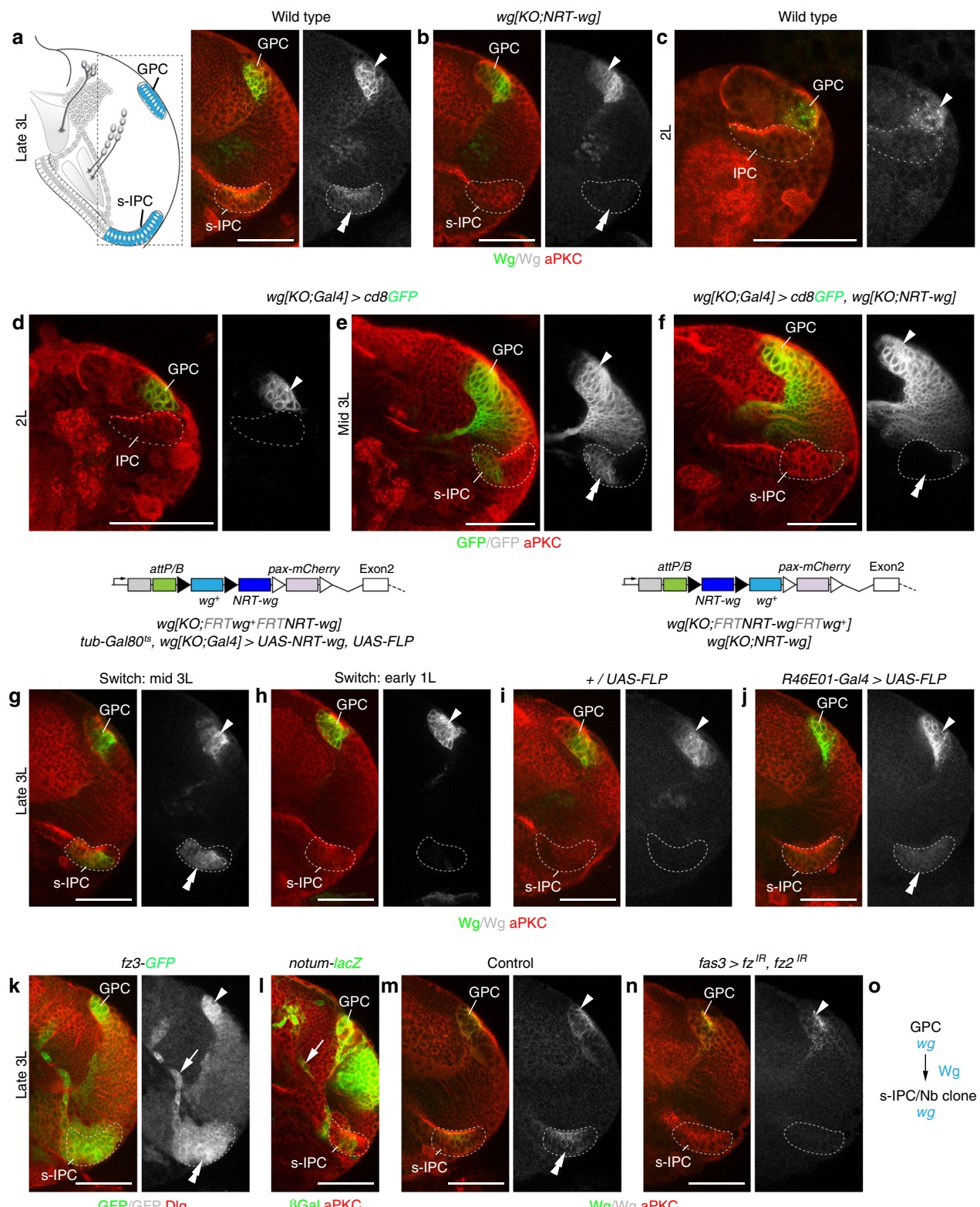

p-IPC, we next conducted lineage-tracing experiments. Because *dpp-Gal4* is expressed in the entire optic lobe, we searched for a driver with activity restricted to the IPC Dpp-expression domains. Expression of *R45H05-Gal4* (Fig. 4a) is controlled by an enhancer fragment of *dorsocross 1*, which encodes a T-box transcription factor and known Dpp target gene[34]. Reporter gene expression was detected in dorsal and ventral p-IPC subdomains, cell streams, Nbs, and postmitotic progeny in larval optic lobes and persisted throughout pupal development into adulthood, where it specifically labeled T4/T5 neurons innervating the layers 3/4 but not 1/2 (Fig. 4b; Supplementary Fig. 4a, b). As the expression was weak, we combined the FLPout approach with *tub-Gal80^{ts}* to permanently label *R45H05-Gal4* expressing progeny, and again solely detected T4/T5 neurons projecting into layers 3/4 (Fig. 4c). Finally, we used the Flybow approach to label lineages in different colors. All clones contained T4/T5 neurons innervating either layers 1/2 or 3/4 but not both (Fig. 4d; *n* = 43 clones in 26 optic lobes), consistent with distinct origins of these neuron subtypes.

In lineage-tracing experiments using *R45H05-Gal4*, clones included C2, the T2-variant T2a, and/or T3 neurons, as judged by their characteristic cell body positions and terminals in the lamina and lobula, respectively (Fig. 4e). While C3, T2, and T3 neurons were not affected, C2 neurons were fully or partially absent in *wg{KO;NRT-wg}* flies with two or three remaining lobula plate layers, respectively (Fig. 4f, g; Supplementary Fig. 4c–j). Due to the lack of specific markers, we could not conduct tests for T2a neurons. Thus, T4/T5 neurons innervating layers 3/4, C2, and possibly T2a neurons are specifically derived from the dorsal and ventral Dpp-expression domains, while the p-IPC core generates T4/T5 for layers 1/2, as well as C3, T2, and T3 neurons (Fig. 4h).

### *dac* and *ato* mediate the transition to T4/T5 neuron formation.

We next asked when and how layer-specific T4/T5 neurons become distinct during development. Dac and Ato are expressed in the second d-IPC Nb competence window[23]. While Dac is initially expressed in all T4/T5 neurons in 3rd instar larvae, albeit with varying expression levels, Dac was only found in ~50% of adult T4/T5 neurons (Fig. 5a–c; see below). The enhancer trap *dac^{p7d23}-Gal4* line faithfully reported Dac expression in T4/T5 neurons throughout development and remained active in neurons innervating layers 1/2 (Fig. 5d, e; Supplementary Fig. 5a). Thus, Dac is downregulated in layer 3/4 innervating T4/T5 neurons and maintained in neurons projecting to layers 1/2 (Fig. 5f).

To examine the function of *dac*, we used mosaic analysis with a repressible cell marker (MARCM) to generate *dac^1* mutant T4/T5 neurons labeled with the *dac* enhancer Gal4 line *R9B10-Gal4* (Supplementary Fig. 5b, c). Some adult *dac* mutant neurons showed T2/T3-like morphologies with neurites extending into the medulla layer M9 and higher, and synaptic terminals in lobula

layers Lo2 and Lo3 (Fig. 5f–j). This suggested that mutant neurons adopted features of neurons born in the first d-IPC Nb competence window. To assess potential redundancy, we performed *dac* and *ato* knockdown experiments using validated *UAS-RNAi* transgenes (Supplementary Fig. 5d–g). In samples with IPC-specific single *dac* or *ato* knockdown, frequently only one Connectin-positive lobula plate layer was discernible (Supplementary Fig. 5h–j). However, simultaneous *dac* and *ato* knockdown caused the absence of neurons with T4/T5 morphologies and in consequence an undersized lobula plate neuropil. Fas3 in T4/T5 dendrites within medulla layer Me10 and lobula layer Lo1 and Connectin in lobula plate layers were severely reduced. Consistent with T2/T3-like morphologies, remaining neurons innervated the medulla and lobula (Fig. 5k–n; Supplementary Fig. 5k).

Therefore, *dac* and *ato* are required together for the switch from T2/T3 to T4/T5 neuron formation. Dac is maintained in layer 1/2 innervating T4/T5 neurons, but downregulated in layer 3/4 innervating T4/T5 neurons, suggesting that layer 1/2 identity could represent the default fate.

### Notch controls the choice between T4 and T5 neuron identity.

In the OPC, Notch signaling in asymmetric GMC divisions contributes to generating neuronal diversity[35], involving differential apoptosis in region-specific lineages[36]. We therefore assessed whether this pathway could mediate the distinction of layer 1/2 and 3/4 innervating T4/T5 neurons downstream of Dpp. We did not detect apoptotic cells in the 3rd instar larval lobula plate (Supplementary Fig. 6a). However, expression of an activated, ligand-independent form of Notch (N^{intra}) in d-IPC Nbs in the second competence window and their progeny using the late *R9B10-Gal4* driver affected T4 neuron formation, because neurites were missing in the medulla of 3rd instar larvae (Fig. 6a, b). Adults exhibited a milder phenotype, in which mostly T4 neurons connecting to the anterior proximal medulla were affected (Supplementary Fig. 6b–h). Conversely, IPC-specific knockdown of the transcriptional regulator Suppressor of Hairless (Su(H)) using *R17B05-Gal4* caused the absence of T5 neurons in adults, whereas T4 neurons were present (Fig. 6c, d; Supplementary Fig. 6f). In these flies, lobula plate layers 3 and 4 could not be discriminated (Supplementary Fig. 6i). The lobula plate and lobula neuropils were severely disorganized, likely because of an early requirement of Notch in p-IPC NE cells during the 3rd instar larval stage[27] (Supplementary Fig. 6j, k). Hence, Notch controls the choice between T4 and T5 neuron fate (Fig. 6e), whereas the distinction between layer 1/2 Dac-positive and layer 3/4 Dac-negative T4/T5 neurons is mediated by a Notch-independent mechanism.

### Dpp-dependent specification of layer 3/4 T4/T5 neurons by *omb*.

While *dpp* reporter gene activity extended from the p-IPC

---

**Fig. 2** The GPC areas release Wg to induce *wg* in the s-IPC. **a**, **b** Schematic in **a** highlights the region of interest shown in subsequent panels. Unlike in controls (**a**), Wg immunolabeling (green) was absent in the s-IPC (dashed line, double arrowhead) in many *wg{KO;NRT-wg}* flies (**b**). The GPC areas (arrowhead) were not affected. **c**, **d** In 2nd instar larvae (2L), Wg protein (**c**, green) and *wg{KO;Gal4} UAS-cd8GFP* (**d**, green) were detected in GPC areas (arrowhead), but not in the adjacent IPC (dashed line). **e**, **f** s-IPC-specific *wg{KO;Gal4} UAS-cd8GFP* expression (green, double arrowheads) in mid 3rd instar larvae (**e**) was absent in *wg{KO;NRT-wg}* flies (**f**). Arrowheads indicate expression in GPC area. **g**, **h** In controls, *wg{KO;Gal4} UAS-FLP* mediated *wg{KO;FRT wg^+ FRT NRT-wg}* allele switching and simultaneous *UAS-NRT-wg* overexpression were induced at the mid 3rd instar larval stage (**g**, *wg^+* background). Allele switching at the 1st instar larval stage (**h**, *NRT-wg* background) did not rescue s-IPC-specific Wg loss (green). **i**, **j** Unlike in controls (**i**), *R46E01-Gal4 UAS-FLP*-mediated GPC areas-specific *wg{KO;FRT NRT-wg FRT wg^+}* allele switching (**j**) rescued s-IPC-specific NRT-Wg loss (green). Filled and open triangles in transgene schematics represent *FRT* and *loxP* sites, respectively (**g–j**). **k**, **l** The Wg target gene reporter lines *fz3^{G00357}-GFP* (**k**, green) and *notum^{WRE}-lacZ* (**l**, green) show expression in the GPC areas, the s-IPC, and in migratory progenitors (arrow) originating from the adjacent p-IPC. **m**, **n** Unlike in controls (**m**), *fas3^{NP1233}-Gal4*-mediated IPC-specific *fz* and *fz2* knockdown (**n**) caused loss of Wg (green) in the s-IPC. **o** Summary of *wg* function in the GPC areas. For genotypes and sample numbers, see Supplementary Table 1. Scale bars, 50 μm

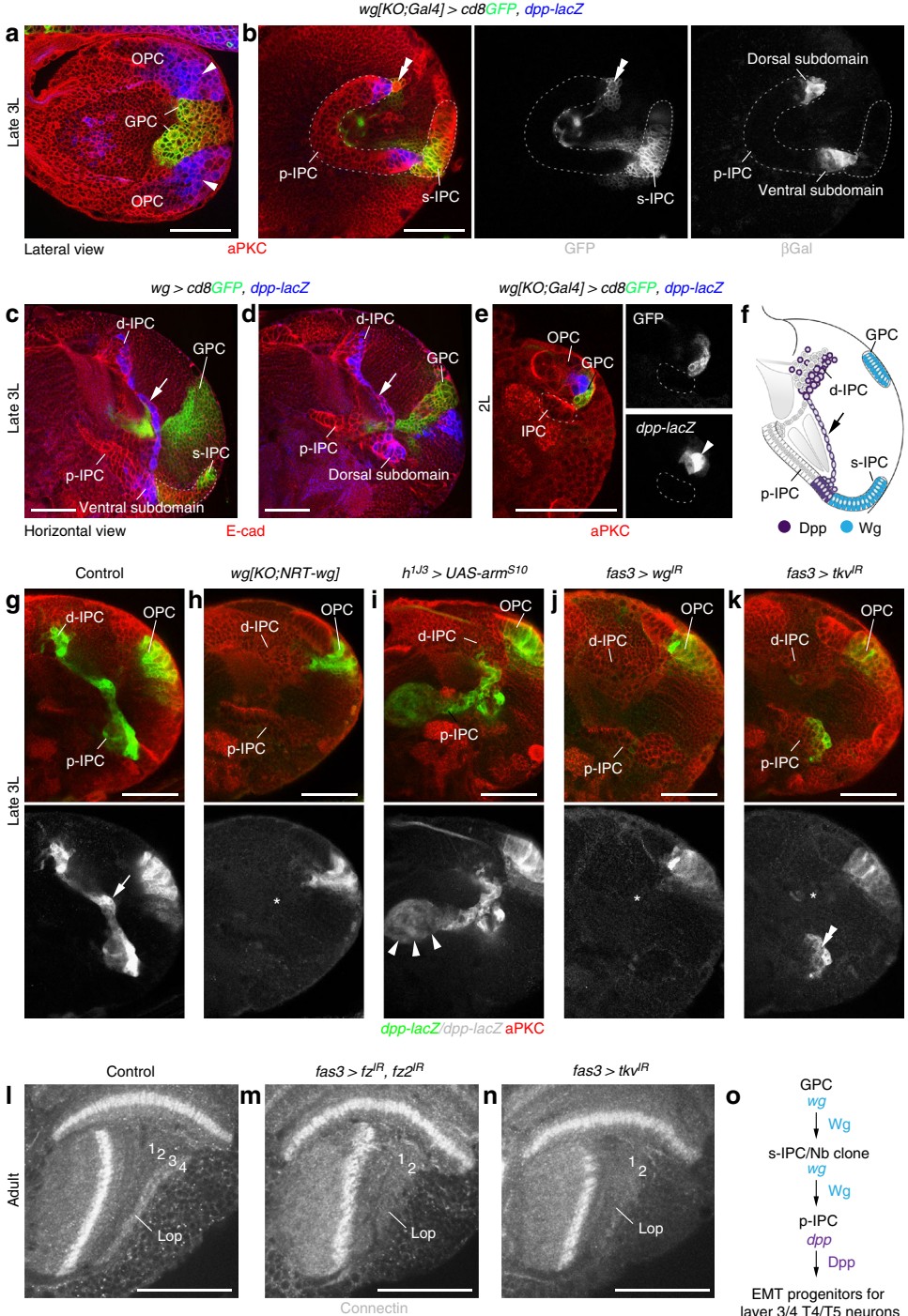

**Fig. 3** s-IPC-derived Wg is required for Dpp-dependent EMT in the p-IPC. **a**, **b** In 3rd instar larvae (3L), *dpp-lacZ* (blue) was detected in subdomains adjacent to *wg{KO;Gal4} UAS-cd8GFP* (green) regions in the dorsal and ventral OPC (**a**, arrowheads) and p-IPC (**b**). Double arrowhead indicates GFP-positive Nb clone adjacent to the dorsal p-IPC. **c**, **d** *dpp-lacZ* (blue) was maintained in progenitor streams (arrows) from the ventral and dorsal p-IPC subdomains. **e** In 2nd instar larvae (2L), *dpp-lacZ* (blue) was present in the OPC (arrowhead) adjacent to *wg{KO;Gal4} UAS-cd8GFP*-positive GPC areas (green), but was absent in the IPC (dashed line). **f** Schematic illustrating Wg and Dpp expression domains. Arrow indicates progenitor stream originating from the ventral p-IPC subdomain. **g–k** Unlike in controls (**g**, arrow), *dpp-lacZ* (green) was absent from the IPC in *wg{KO;NRT-wg}* flies (**h**, asterisk). The OPC was not affected. *dpp-lacZ* was ectopically induced in the IPC (arrowheads) by *h^{1J3}-Gal4*-mediated expression of *UAS-arm^{S10}* (**i**). *dpp-lacZ* was absent following *fas3^{NP1233}-Gal4*-mediated IPC-specific *wg* knockdown. In the Dpp-expression domain, this caused EMT defects and loss of progenitor streams (**j**, asterisks). Similar defects were caused by IPC-specific *tkv* knockdown. *dpp-lacZ* remains expressed in the p-IPC (**k**, double arrowheads).
**l–n** Compared to controls (**l**), *fas3^{NP1233}-Gal4*-mediated IPC-specific knockdown of *fz* and *fz2* (**m**) and *tkv* (**n**) caused the loss of lobula plate (Lop) layers 3/4 labeled with Connectin in adults. **o** Summary of *wg* and *dpp* function in the GPC areas, the s-IPC/Nb clone, and p-IPC. For genotypes and sample numbers, see Supplementary Table 1. Scale bars, 50 μm

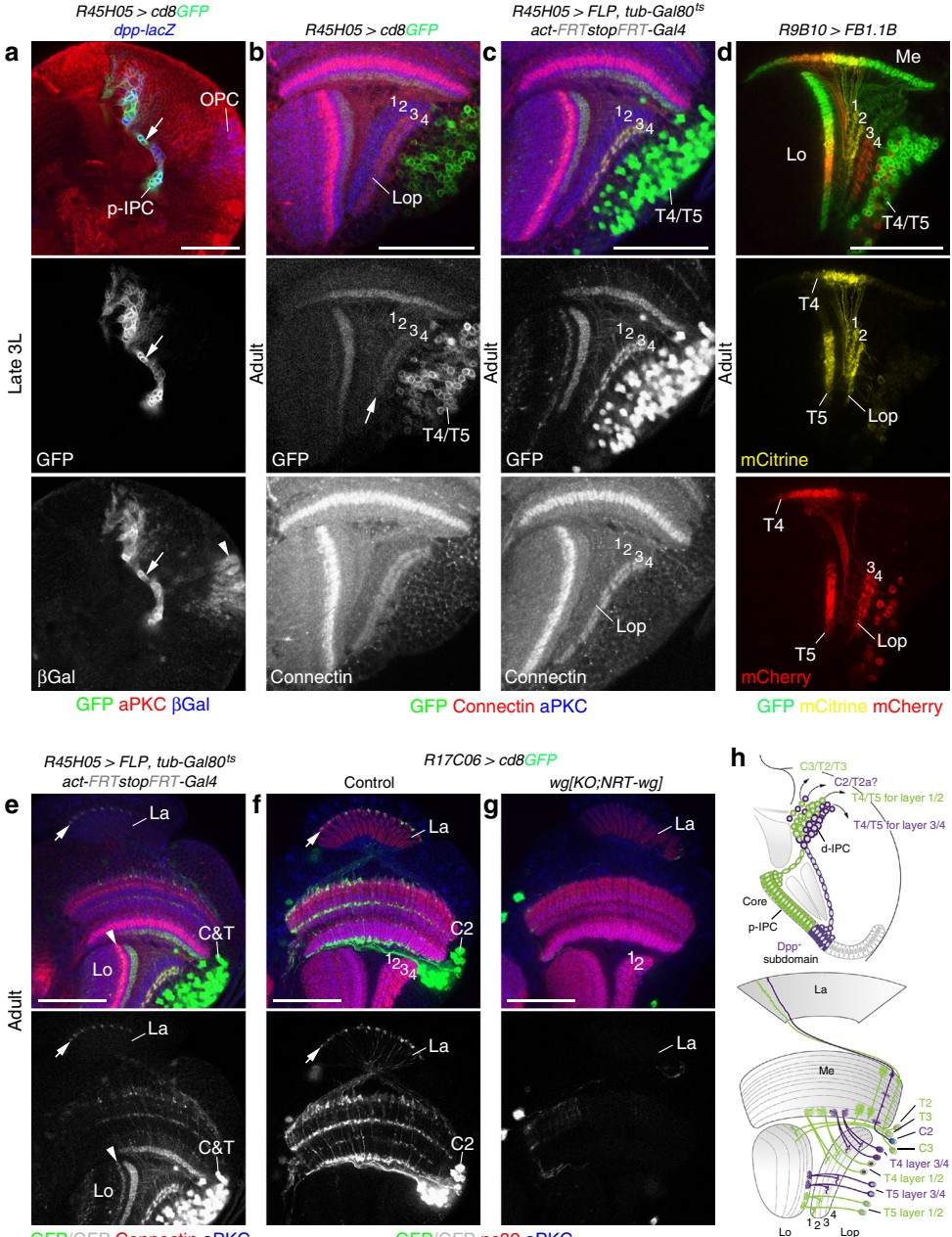

**Fig. 4** The Dpp-expression domain generates C2 and T4/T5 neurons for layers 3/4. **a** *R45H05-Gal4 UAS-cd8GFP* (green) colocalizes with the *dpp-lacZ* (blue) expression domain in the IPC (arrow) but not the OPC (arrowhead) in 3rd instar larvae (3L). **b** *R45H05-Gal4 UAS-cd8GFP* (green) was weakly maintained in T4/T5 neurons innervating Connectin-positive lobula plate (Lop) layers 3/4 (red, arrow) in adults. **c** Permanent GFP-labeling (green) of the *R45H05-Gal4* expression domain using *act>y⁺>Gal4 UAS-GFP, UAS-FLP,* and *tub-Gal80ᵗˢ* was specific to T4/T5 neurons innervating lobula plate layers 3/4. **d** *R9B10-Gal4* in conjunction with Flybow transgenes labeled T4/T5 neuron clones either innervating lobula plate (Lop) layers 1/2 (mCitrine, yellow) or 3/4 (mCherry, red). **e** Permanent GFP labeling (green) with *R45H05-Gal4* included C2, as well as T2a and/or T3 neurons with characteristic axon terminals in the lamina (La, arrow) and lobula (Lo, arrowhead). **f, g** Unlike in controls (**f**), C2 neuron-specific *R17C06-Gal4 UAS-cd8GFP* expression (green) with axon terminals (arrow) in the lamina (La) was absent in *wg{KO;NRT-wg}* flies that had two remaining lobula plate layers (**g**). **h** Schematics illustrating the neuron subtypes derived from the Dpp-expression domain and the core p-IPC in 3rd instar larvae and adults. For genotypes and sample numbers, see Supplementary Table 1. Scale bars, 50 μm

to the d-IPC, phospho-Mad (pMad) labeling was restricted to p-IPC NE cells (Fig. 7a). Consistently, knockdown experiments using Gal4 lines with progressively restricted activities (Supplementary Fig. 7a–c) revealed a requirement of *tkv* for layer 3/4 neuron formation in p-IPC NE cells (cf. Fig. 3n), but not in d-IPC Nbs and T4/T5 neurons (Supplementary Fig. 7d–f). Therefore, an additional mechanism must relay Dpp signaling activity in p-IPC

NE cells to distant postmitotic T4/T5 neurons. The T-box transcription factor Optomotor blind (Omb) is a Dpp target in the p-IPC[23]. In 3rd instar larval brains, Omb expression is maintained in progenitors, Nbs, and T4/T5 neurons derived from the Dpp-positive p-IPC subdomains (Fig. 7b, c). Expression persisted in adult T4/T5 neuron subsets (Supplementary Fig. 7g). In *wg{KO;NRT-wg}* flies, *omb* expression was severely reduced in

the p-IPC and progeny, but not in the OPC (Fig. 7d, e). Consistently, Omb was also decreased following IPC-specific *tkv* knockdown (Fig. 7f, g).

To determine the function of *omb*, we conducted knockdown experiments using *fas3-Gal4* and two validated *UAS-RNAi* transgenes (Supplementary Fig. 7h–k). Experimental animals were raised at 18 °C and shifted to 29 °C at the early 3rd instar larval stage, because *omb* is expressed in the embryonic optic lobe

placode and null mutations cause severe disorganization or complete loss of adult optic lobes[37]. Under these conditions, *omb* knockdown did neither affect *dpp* expression nor the EMT of progenitors from the p-IPC (Fig. 7h, i). However in adults, lobula plate layers 3/4 were absent (Fig. 7j, k). In contrast to *tkv*, *omb* knockdown using Gal4 lines with progressively restricted activities revealed that this transcription factor is required in Nb in the second competence window and postmitotic T4/T5

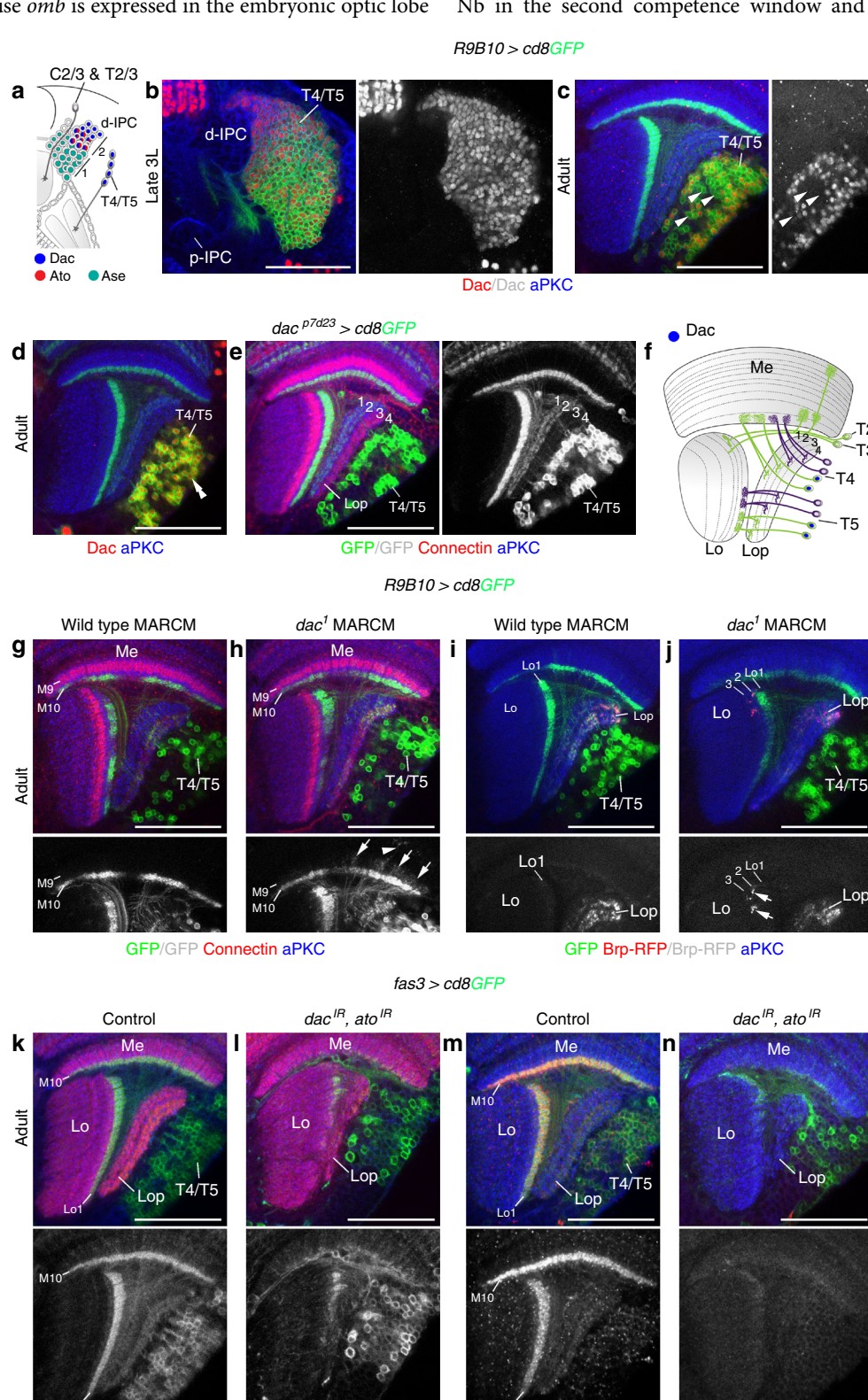

neurons for layer 3/4 neuron specification (Fig. 7l, m). Thus, *omb* could relay subdomain-specific Dpp signaling effects in p-IPC NE cells across intermediate cellular states, i.e., migratory progenitors, Nbs, and T4/T5 progeny (Fig. 7n).

**omb mediates layer 3/4 T4/T5 neuron specification by Dac repression**. Finally, we assessed whether Omb could be responsible for downregulating Dac. In 3rd instar larvae, Omb was co-expressed with high levels of Dac in younger and low levels in older T4/T5 neurons (Fig. 8a). From 24 h after puparium formation (APF) (Fig. 8b) to adulthood (Supplementary Fig. 8a), Omb and Dac showed mutually exclusive expression in T4/T5 neurons. While Dac was maintained in ~50% of adult T4/T5 neurons in controls (Fig. 8c, h), 95% of T4/T5 neurons expressed Dac and adopted layer 1/2 identity following *omb* knockdown (Figs. 7m, 8d, h). Conversely, when *omb* was overexpressed, none of the T4/T5 neurons expressed Dac and all acquired layer 3/4 identity (Fig. 8e, h, j). Under both conditions, many T4/T5 neurons underwent apoptosis in 3rd instar larvae (Supplementary Fig. 8b–d). Consistently in adults, their numbers were reduced by 33% and 48%, respectively (Fig. 8h), possibly because excessive T4/T5 neurons either for layers 1/2 or 3/4 compete for limited trophic support. To demonstrate that *omb* is sufficient, we took advantage of *wg{KO;NRT-wg}* flies, in which all T4/T5 neurons adopted layer 1/2 identity and Dac was expressed in 97% of these (Fig. 8f, h; Supplementary Fig. 8e, f). In this background, *omb* overexpression only mildly affected neuron numbers (19% reduction) and Dac was downregulated (Fig. 8g, h). Importantly, concomitant upregulation of Connectin suggested that these differentiated into layer 3/4 innervating T4/T5 neurons (Fig. 8i, k). Thus, *omb* is required and sufficient for specifying T4/T5 neurons innervating layers 3/4 by downregulating Dac (Fig. 8l).

## Discussion

The spread of Wg is dispensable for patterning of many tissues[30]. However, our study uncovered a distinct requirement for diffusible Wg in the nervous system, where it orchestrates the formation of T4/T5 neurons innervating lobula plate layers 3/4. Their generation depends on inductive mechanisms (Fig. 8m) that are relayed in space and time. The spatial relay consists of a multistep-signaling cascade across several NE domains: Wg from the GPC areas induces *wg* expression in the s-IPC and Nb lineage adjacent to ventral and dorsal p-IPC subdomains; this secondary Wg source activates *dpp* expression. Dpp signaling mediates EMT of migratory progenitors from these subdomains. The p-IPC core produces Dac-positive layer 1/2 specific T4/T5 neurons. Dpp signaling in p-IPC NE subdomains triggers a temporal relay across intermediate cellular states by inducing *omb*. Omb in turn suppresses Dac, conferring layer 3/4 identity to postmitotic T4/T5 neurons.

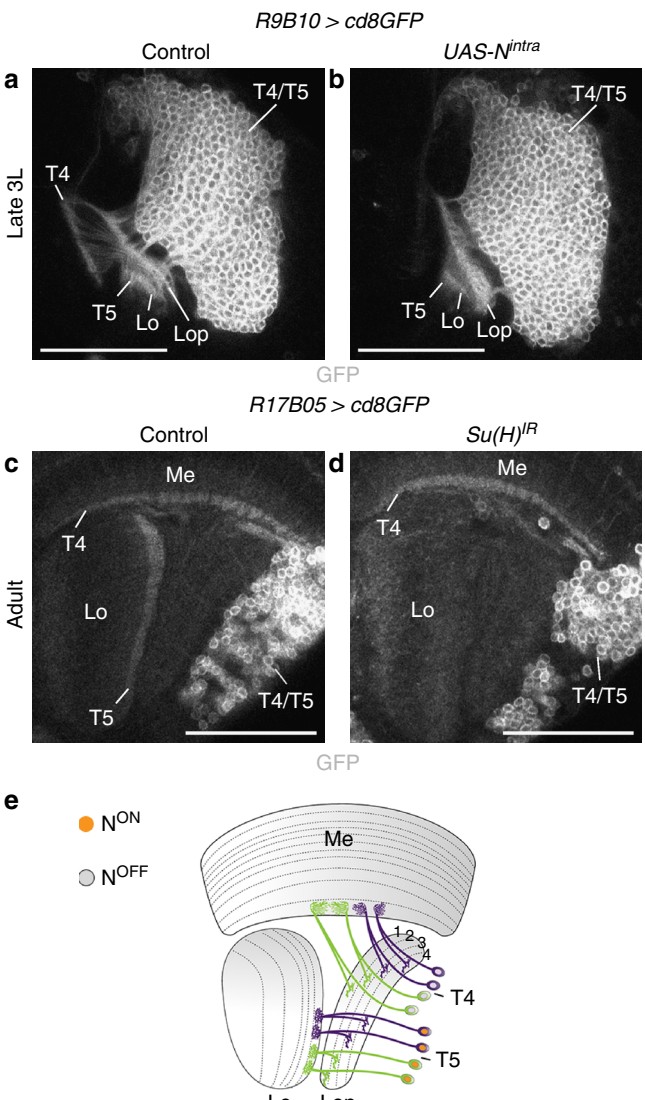

R9B10 > cd8GFP

Control UAS-N^intra

GFP

R17B05 > cd8GFP

Control Su(H)^IR

GFP

**Fig. 6** Notch controls the choice between T4 and T5 neuron fate. **a**, **b** Unlike in controls (**a**), *R9B10-Gal4 UAS-cd8GFP* labeled T4 neurites were absent in 3rd instar larvae following *UAS-N^intra* overexpression in d-IPC Nbs and their progeny during the second competence window (**b**). T5 neurites in the lobula (Lo) were unaffected. **c**, **d** Unlike in controls (**c**), following *R17B05-Gal4 UAS-cd8GFP*-mediated IPC-specific *Su(H)* knockdown, T5 neurites were absent in the adult lobula (**d**). T4 neurites were present. **e** Schematic illustrating the Notch-dependent specification of T5 neurons. For genotypes and sample numbers, see Supplementary Table 1. Scale bars, 50 µm

**Fig. 5** *dac* and *ato* are required for T4/T5 neuron formation. **a** Schematic illustrating the expression of Ase (turquoise), Ato (red), and Dac (blue) in d-IPC Nbs during the first (1) and second (2) competence windows and their progeny. **b**, **c** Dac (red) was expressed in all *R9B10-Gal4 UAS-cd8GFP* (green) labeled T4/T5 neurons in 3rd instar larvae (**b**). It was downregulated (arrowheads) in approximately 50% of adult T4/T5 neurons (**c**). **d**, **e** *dac^{p7d23}-Gal4 UAS-cd8GFP* (green) faithfully reported Dac (red, double arrowhead) expression in adults (**d**), specifically labeling T4/T5 neurons innervating lobula plate (Lop) layers 1 and 2 (**e**). **f** Schematic illustrating Dac expression (blue) in adult T4/T5 neurons. **g**–**j** Unlike in controls (**g**, **i**), *dac^1* mutant T4/T5 neurons adopted T2/T3 neuron morphologies with *R9B10-Gal4 UAS-cd8GFP* (green), displaying neurite extensions into medulla (Me) layer M9 (arrows) and more distal layers (arrowhead) (**h**), and *UAS-brp-RFP* labeled synaptic terminals (red, arrows) in lobula (Lo) layers 2 and 3 (**j**). **k**, **l** Unlike in controls (**k**), *fas3^{NP1233}-Gal4 UAS-cd8GFP* (green)-mediated IPC-specific simultaneous knockdown of *dac* and *ato* generated neurons that failed to form a four-layered lobula plate (Lop) neuropil and dendrites in medulla (Me) layer 10 and lobula (Lo) layer 1 (**l**). **m**, **n** Unlike in controls (**m**), Fas3-positive (red) T4/T5 neurons were absent following IPC-specific simultaneous knockdown of *dac* and *ato* mediated by *fas3^{NP1233}-Gal4 UAS-cd8GFP* (green) (**n**). For genotypes and sample numbers, see Supplementary Table 1. Scale bars, 50 µm

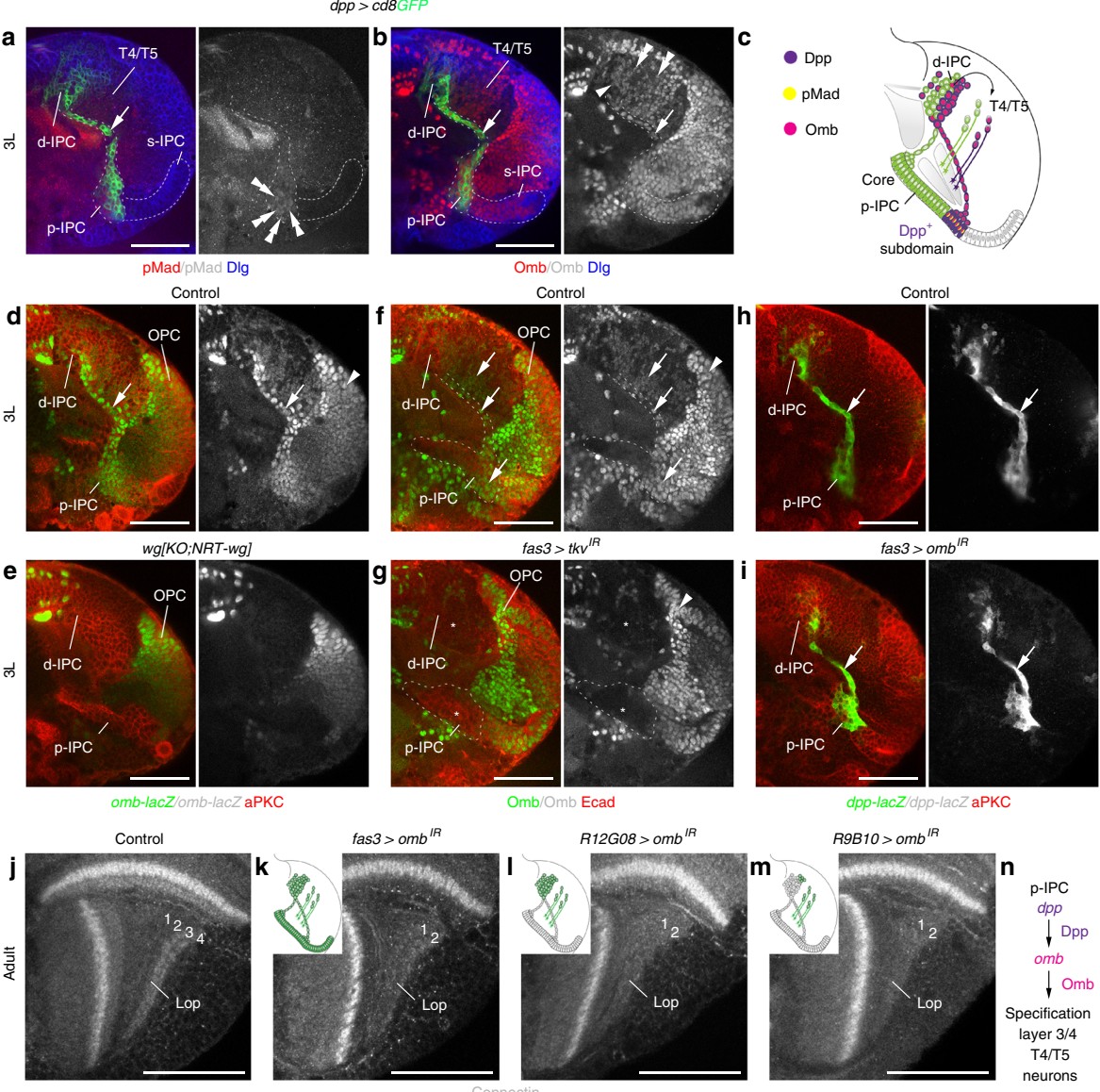

**Fig. 7** Omb mediates Dpp-dependent specification of T4/T5 neurons for layers 3/4. **a** Phosphorylated Mad (pMad, red) in the *dpp-Gal4 UAS-cd8GFP* (green) expression domain was found in p-IPC NE cells (double arrowheads), but not in progenitors (arrow), d-IPC Nbs/GMCs, or T4/T5 neurons. **b** Omb (red) was maintained in the *dpp-Gal4 UAS-cd8GFP* (green) expression domain in progenitors (arrow), d-IPC Nbs/GMCs (arrowhead), and T4/T5 neuron subsets (double arrowheads). **c** Schematic illustrating pMad and Omb distribution within the Dpp expression domain. **d**, **e** Unlike in controls (**d**), *omb[P1]-lacZ* (green) was absent from the IPC (arrow) in *wg{KO;NRT-wg}* flies (**e**). OPC expression (arrowhead) was not affected. **f**, **g** In controls (**f**), Omb protein (green) was detected in p-IPC subdomains, migratory progenitors, and progeny (arrows). Following *fas3[NP1233]-Gal4*-mediated IPC-specific *tkv* knockdown (**g**), expression was severely reduced in these cells (asterisks). OPC expression (arrowhead) was not affected. **h**, **i** Indistinguishable from controls (**h**), *fas3[NP1233]-Gal4*-mediated IPC-specific knockdown of *omb* (**i**) did not affect EMT of *dpp-lacZ*-labeled progenitors (green, arrow) in the IPC. **j–m** Unlike in controls (**j**), *omb* knockdown in the entire IPC and its progeny using *fas3[NP1233]-Gal4* (**k**), in the d-IPC and postmitotic T4/T5 neurons using *R12G08-Gal4* (**l**), and primarily postmitotic T4/T5 neurons using *R9B10-Gal4* (**m**) resulted in the absence of Connectin-positive lobula plate layers 3/4. Schematic insets highlight cell type-specificities of Gal4 lines in green. **n** Schematics summarizing the role of Dpp in inducing Omb expression to specify layer 3/4 innervating T4/T5 neurons. For genotypes and sample numbers, see Supplementary Table 1. Scale bars, 50 μm

When Wg is membrane-tethered, the first step of this cascade is disrupted. This defect is not caused by decreased signaling activity of NRT-Wg protein in *wg{KO;NRT-wg}* flies. First, wild-type Wg signaling activity inside the GPC areas and the adjacent OPC was not affected. Second, in allele switching experiments, ectopic expression of a highly active *UAS-NRT-wg* transgene in the GPC areas was unable to rescue. By contrast, restoring wild-type *wg* function in the GPC areas was able to rescue, supporting the notion that Wg release and spread from

the GPC areas are required to induce its own expression in the s-IPC and the Nb clone.

Although Wg release is essential, the range of action is likely limited. Wg expression in the s-IPC commences in early 3rd instar larvae, when it is still in close proximity with the GPC. Half of the *wg{KO;NRT-wg}* flies showed residual *dpp* expression in one progenitor stream at the 3rd instar larval stage and a 25% reduction of T4/T5 neurons, correlating with three lobula plate layers in adults. The other half lacked *dpp-lacZ* expression and

showed a 50% reduction of T4/T5 neurons correlating with two remaining layers. While this partial phenotypic penetrance is not fully understood, NRT-Wg likely partially substituted for Wg because of the initial close proximity of the GPC areas and the s-IPC and Nb clone. Occasional residual NRT-Wg expression in the s-IPC argues against an all-or-nothing inductive event and

suggests a model, whereby cell-intrinsic signaling thresholds have to be reached. Theoretically, the *dpp* expression defect in the p-IPC of *wg{KO;NRT-wg}* flies could reflect the dependence on long-range Wg from the GPC areas. However, as we have shown, IPC-specific *wg* knockdown leads to *dpp* loss in the p-IPC. Propagation of sequential Wnt signaling could explain long-range

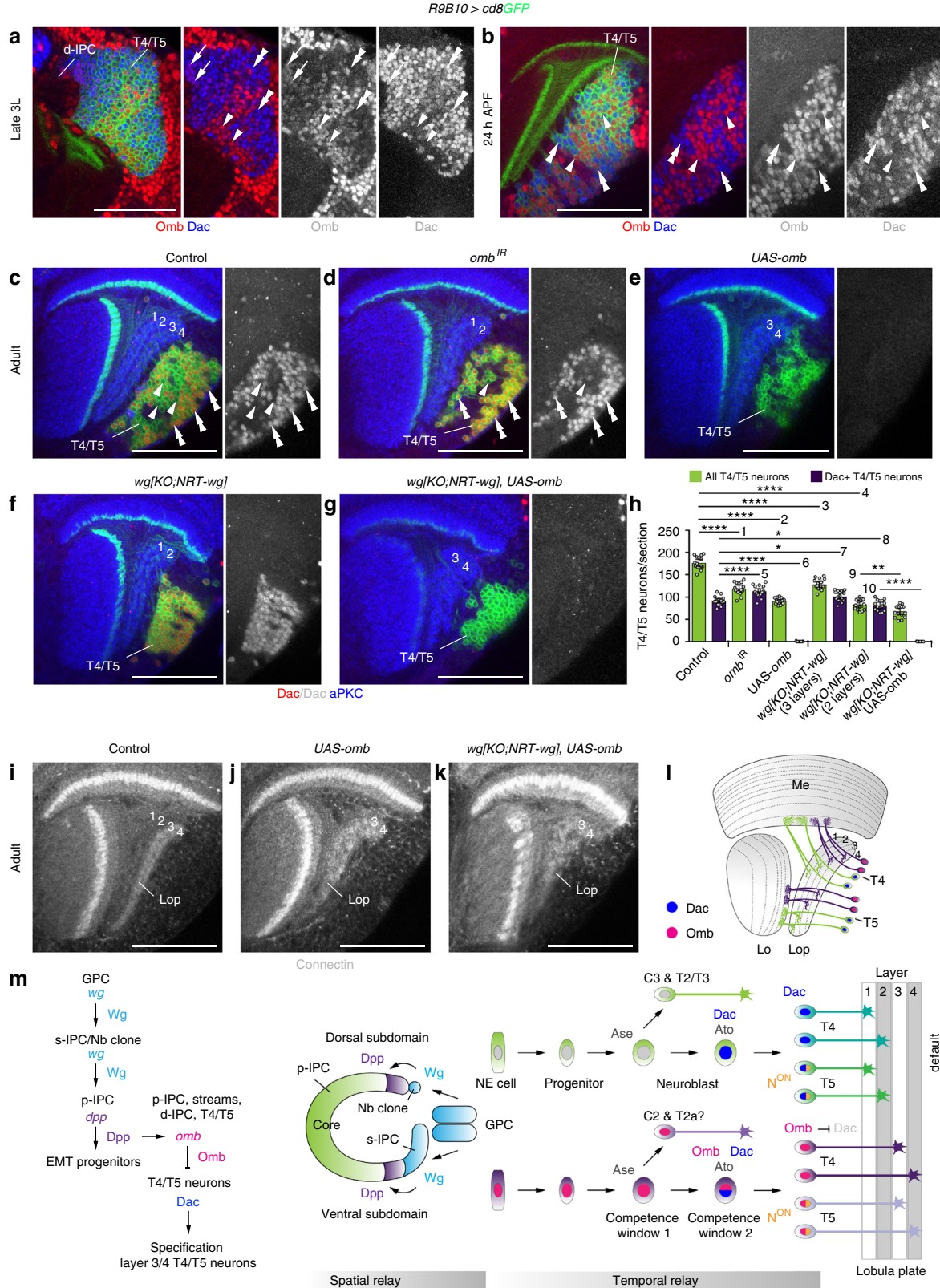

activities[38,39]. Moreover, sequentially acting primary and secondary sources of Wg have been described in the developing *Drosophila* eye[40], suggesting that the regulatory mechanism observed in the optic lobe might be employed in several contexts. The different outcomes of early and late allele *wg* to *NRT-wg* allele switching indicate that Wg secretion is required for the induction but not long-term maintenance of *wg* expression in the s-IPC. The GPC areas become rapidly separated from the s-IPC and Nb clone by compact rows of newly generated neurons. As part of a relay system, diffusible Wg may therefore be required to bridge distances over a few cell diameters during the initial phase of neurogenesis. The s-IPC in *wg{KO;NRT-wg}* flies expressed Hth and generated two neuron clusters as in wild-type. Thus, the sole function of *wg* in the s-IPC is to relay the GPC-derived Wg signal to induce *dpp* expression in the p-IPC. Since Wg release is not required in the GPC areas to induce *dpp* in the adjacent OPC, this secondary *wg* function in the s-IPC is most likely juxtacrine.

Compared to approximately 80 medulla neuron subtypes derived from the OPC[36], the specification of 13 distinct subtypes originating from the p-IPC appears simple. However, the distinct mechanisms employed are surprisingly complex. We previously showed that cross-regulatory interactions between *D* and *tll* regulate a Nb competence switch from generating early-born C2, C3, T2, T2a, and T3 neurons to eight distinct layer-specific T4/T5 subtypes. Ato and Dac are expressed in the second Nb competence window and depend on *tll*[23]. Our functional studies showed that *dac* mutant T4/T5 neurons adopted early-born T2/T3 neuron-like morphologies. Similarly, *ato* mutant T4/T5 neurons displayed neurite connectivity defects[41]. Notably, simultaneous knockdown of *dac* and *ato* resulted in the absence of T4/T5 neurons, demonstrating that both are required together for the ability of d-IPC Nbs to produce new neuron subtypes in the second competence window.

Dac is initially expressed in all T4/T5 neurons but only maintained in layer 1/2 innervating subtypes. This suggests that an essential step for the specification of layer 3/4 innervating neurons is the downregulation of Dac and the suppression of the T4/T5 default neuron fate, i.e., layer 1/2 identity. Although the mode of this inhibitory mechanism depends on the outcome of the Nb-specific switching mechanism in the d-IPC, it is already primed in p-IPC NE cells. Thus, layer-specificity and therefore motion-directionality are determined early in the NE precursors of T4/T5 neurons. Molecularly, it involves the Omb-mediated relay of Dpp-signaling-dependent NE cell patterning information across intermediate cell states to postmitotic T4/T5 neurons resulting in the repression of Dac. In contrast to the OPC[36], we found no link between NE patterning in the p-IPC and Notch-dependent differential apoptosis of region-specific

T4/T5 subtypes. Instead, Notch controls the choice between T4 and T5 identity, likely during the second competence window, indicating that the distinction between layer 1/2 and 3/4 fates precedes T4 and T5 neuron specification.

The mechanisms controlling the maintenance of *omb* expression, and Omb-mediated downregulation of Dac are unclear. Hypotheses regarding the latter have to be reconciled with the fact that *dac*, together with *ato*, is required for the formation of all T4/T5 neurons and hence is expressed in all d-IPC Nbs during the second competence window. Omb and Dac are initially co-expressed in Nbs and young T4/T5 neurons, suggesting that Omb does not directly repress *dac* transcription. Yet, expression of the *dac^{p7d23}* enhancer trap Gal4 line showed that *dac* is only transcribed in layer 1/2 neurons in adults. A possible scenario is that Omb could break Dac autoregulation by triggering degradation of Dac. Since T-box genes can act as transcriptional activators and repressors[42] and their effects are influenced by various co-factors[43], future studies will need to explore the molecular details underlying Omb-mediated repression of Dac. It will also be important to determine whether layer 3/4 specification is mediated solely by Dac downregulation, or whether *omb* has additional instructive roles.

Consistent with the observation that C2 and C3 neurons have distinct developmental origins[44], we found that Nbs derived from the Dpp-expression domain produce C2 and possibly T2a neurons during the first Nb competence window, while the core p-IPC generates C3, T2, and T3 neurons. *dac* mutant T4/T5 neurons adopt T2/T3-like morphologies suggesting that this is the default neuron fate in this neuron group. While Omb is maintained in C&T neurons derived from the Dpp-expression domain, Dac is not expressed, suggesting that Omb interacts with other molecular determinants in these neurons. While we did not explore how layer 1 and 2 neurons or layer 3 and 4 neurons become distinct from each other because of the lack of specific markers, our data suggest a possible contribution of Ato/Dac and Notch signaling, as these are active within the d-IPC. Findings in a concurrent study of Pinto-Teixeira et al.[45] align with our data concerning the role of Dpp and Notch signaling. Furthermore, a second study of Mora et al.[46] reported an additional role for Ato in controlling the transient amplification of d-IPC Nbs by symmetric cell division to ensure that the correct number of T4/T5 neurons is produced. It will be fascinating to identify the transcriptional targets of Notch, Ato/Dac, and Omb that mediate ganglion- and layer-specific targeting of T4/T5 dendrites and axons, respectively. Finally, future behavioral studies of layer 3/4-deficient flies will address to what extent direction selectivity is affected or compensatory mechanisms are in place.

**Fig. 8** Omb converts layer 1/2 into layer 3/4 T4/T5 neurons by Dac downregulation. **a, b** In the 3rd instar larvae (**a**), Omb (red) and Dac (blue) were initially co-expressed (arrows) in new-born *R9B10-Gal4 UAS-cd8GFP* (green) labeled T4/T5 neurons close to the d-IPC. In some more mature T4/T5 neurons, that were positive for Omb, Dac expression was low (arrowheads), while in others, Dac expression was high (double arrowheads) and Omb expression was absent. At 24 h after puparium formation (APF) (**b**), Dac (double arrowheads) and Omb (arrowheads) show mutually exclusive expression. **c–g** In controls (**c**), Dac (red, double arrowheads) was expressed in approximately 50% of T4/T5 neurons. Arrowheads indicate Dac-negative *R9B10-Gal4 UAS-cd8GFP* (green) labeled T4/T5 neurons. Dac was maintained in almost all T4/T5 neurons following *omb* knockdown (**d**), and downregulated following *omb* overexpression (**e**). Dac was expressed in all T4/T5 neurons in *wg{KO;NRT-wg}* flies with two lobula plate layers (**f**). Ectopic Omb was sufficient to downregulate Dac in these flies (**g**). **h** Quantification of all and Dac-positive T4/T5 neuron numbers following *omb* manipulations. The scatter plot with bars shows data points and means with ±95% confidence interval error bars ($n = 15$ corresponding to three serial optical sections, 6-μm apart, from five samples per genotype). Unpaired, two-tailed Student's *t*-test not assuming equal variance: $P = 5.57 \times 10^{-12}$, $P = 6.19 \times 10^{-15}$, $P = 1.26 \times 10^{-10}$, $P = 1.98 \times 10^{-17}$, $P = 3.60 \times 10^{-5}$, $P = 4.79 \times 10^{-15}$, $P = 0.020$, $P = 0.017$, $P = 0.0015$, $P = 2.42 \times 10^{-13}$. *$P < 0.05$; **$P < 0.01$; **** $P < 0.0001$. **i–k** Unlike in controls (**i**), *R9B10-Gal4*-mediated ectopic *UAS-omb* expression in T4/T5 neurons of wild type (**j**) or *wg{KO;NRT-wg}* (**k**) flies resulted in ectopic Connectin expression in the lobula plate (Lop). **l** Schematic illustrating Dac and Omb expression in adults. **m** Working model for spatial and temporal relay mechanisms regulating the formation and specification of layer-specific T4/T5 neurons. For genotypes and sample numbers, see Supplementary Table 1. Scale bars, 50 μm

Signaling centers, also called organizers, pattern tissues in a non-autonomous fashion[47]. The vertebrate roof plate and the cortical hem, for instance, both release Wnts and Bmps to pattern NE cells in the developing dorsal spinal cord and in the surrounding forebrain, respectively[47–49]. In the *Drosophila* visual system, the GPC areas express *wg* and pattern the OPC by inducing *dpp* expression in adjacent dorsal and ventral OPC subdomains[26]. Together with our insights into the function of GPC-derived *wg* in IPC patterning and neurogenesis, this firmly establishes the GPC areas as local organizers of optic lobe development. At the onset of neurogenesis, *wg* is first expressed in the GPC areas followed by the s-IPC, explaining the well-established delay in neurogenesis between the IPC and OPC[20]. Wg release from the GPC areas could coordinate the timely onset of neurogenesis in the OPC and IPC to safeguard the alignment of matching partner neurons across several retinotopically organized neuropils. The intercalation of new-born neurons between both neuroepithelia may have driven the need for a relay system using primary and secondary sources of Wg. Wg induces Dpp to subdivide the adjacent OPC and p-IPC NE into distinct regions as basis for generating neuronal diversity. The temporal relay mediated by Omb represents an efficient strategy to pass the memory of spatial NE patterning information by Dpp signaling on to postmitotic neurons generated at a distance. It is thus intricately tuned to the distinct neurogenesis mode of the p-IPC essential for spatially matching birth-order-dependent neurogenesis between the OPC and IPC[23]. Interestingly, the progressive refinement of NE patterning by the induction of secondary signaling centers plays a central role in vertebrate brain development[47]. Furthermore, similar signaling cascades have been recently identified in mammalian optic tissue cultures where sequential Wnt and Bmp signaling induces the expression of the Omb-related T-box transcription factor Tbx5 to specify dorsal retinal NE cells[50]. Hence, such cascades could represent conserved regulatory modules that are employed repeatedly during invertebrate and vertebrate nervous system development.

## Methods

**Genetics**. *Drosophila melanogaster* strains were maintained on standard medium at 25 °C except for Gal80[ts] and RNAi experiments, in which progeny were shifted from 18 °C or 25 °C to 29 °C at specific time points as indicated below; *w1118* flies were used as controls. If not otherwise indicated, all stocks were obtained from the Bloomington *Drosophila* Stock Center or the Vienna *Drosophila* Resource Center and are described in FlyBase. Crosses involved approximately five males and seven virgin females. To avoid overcrowding, parents were transferred to fresh vials every day or every second day. Control and experimental animals of the correct age and genotype were selected irrespective of their gender randomly and independently from several vials. If not otherwise indicated, wandering 3rd instar larvae were analyzed. Detailed genotypes of all experimental samples are listed in Supplementary Tables 1 and 2.

The following stocks served as reporter lines: (1) *fz3*-GFP (Flytrap#G00357)[51], (2) *dpp-lacZ^Exel.2*, (3) *omb^P1*-lacZ[52], (4) *notum^WRE*-lacZ (from J. P. Vincent, The Francis Crick Institute, London). The following lines were used as Gal4 drivers: (1) *dac^p7d23*-Gal4/CyO[53], (2) *dpp^blk1*-Gal4/CyO (from J.P. Vincent), (3) *fas3^NP1233*-Gal4 (Kyoto *Drosophila* Stock Center), (4) *fkh*-Gal4 (from A. Gould, The Francis Crick Institute, London), (5) *h^1J3*-Gal4 (from A. Brand, The Gurdon Institute, Cambridge), (6) *R9B10*-Gal4, (7) *R9H07*-Gal4, (8) *R12G08*-Gal4, (9) *R17B05*-Gal4, (10) *R17C06*-Gal4, (11) *R34E01*-Gal4, (12) *R45H05*-Gal4, (13) *R46E01*-Gal4, (14) *R67E05*-Gal4[54–56], (15) *wg*-Gal4/CyO, (16) *wg{KO;Gal4}/CyO*[30], (17) *wg{KO;Gal4}/CyO; UAS-HRP-cd8GFP/TM2*[30]. For presynaptic labeling, *UAS-brp-RFP* (from S. Sigrist, FU Berlin) was used; for membrane-tethered GFP reporter gene expression, (1) *UAS-cd8GFP*, (2) *UAS-FB1.1B*[57], or (3) *UAS-FB1.1C*. In *UAS-FB1.1C*, cd8-mCherry and myr/palm (mp)-tethered mTurquoise in *UAS-FB1.1B* were replaced by TagRFP-T and mTurquoise2, respectively (unpublished). For *R45H05*-Gal4 cell lineage analysis using the *hs*-FLPout approach[58], *act>y^+>Gal4 UAS-GFP/GlaBc; UAS-FLP/TM6B* flies were crossed with *tubP-Gal80^ts; R45H05*-Gal4 and kept at 18 °C before shifting to 29 °C at the late 3rd instar larval stage. For Flybow experiments[57,59], (1) *hs-mFLP5^MH12/GlaBc* flies were crossed with (2) *UAS-FB1.1B^260b; R9B10*-Gal4 and (3) *UAS-FB1.1B^260b; R9H07*-Gal4. Progeny were heat shocked for 40–60 min at 48–72 h and 72–96 h or at 24–48 h, 48–72 h, and 72–96 h after egg laying (AEL) in a 37 °C water bath. Loss-of-function analysis using mosaic analysis with a repressible cell marker (MARCM)[60] were conducted by crossing *yw*

*hs-FLP122; tubP-Gal80^LL10 FRT40A/CyO; UAS-FB1.1B^49b R9B10-Gal4/TM6* flies with (1) *FRT40A*, (2) *dac^1 FRT40A*, (3) *FRT40A; UAS-brp-RFP*, and (4) *dac^1 FRT40A; UAS-brp-RFP* (*dac^1* from F. Pignoni, SUNY Upstate Medical University). Twenty-four-hour embryo collections were heat shocked for 40–60 min at 48–72 h and 72–96 h AEL in a 37 °C water bath. For knockdown experiments using UAS-RNAi transgenes, the following driver lines were used: (1) *yw ey^3.5-Gal80; fas3^NP1233-Gal4; UAS-Dcr2 UAS-cd8GFP/TM6B*, (2) *R17B05-Gal4 UAS-FB1.1C^49B*, (3) *UAS-cd8GFP; R12G08-Gal4*, and (4) *UAS-FB1.1B^260b; R9B10-Gal4*. RNAi lines used were: (1) *UAS-ato^IR TRiP.JF02089*, (2) *UAS-dac^IR KK106040*, (3) *UAS-fz^IR GD43077*, (4) *UAS-fz2^IR KK108998*, (5) *UAS-omb^IR Cl* (ref. [61]; from G. Pflugfelder, University of Mainz), (6) *UAS-omb^IR KK100598*, (7) *UAS-Su(H)^IR KK103597*, (8) *UAS-tkv^IR KK105834*, and (9) *UAS-wg^IR GD13351*. *UAS-Dcr2* was co-expressed in all experiments. Progeny were shifted from 25 °C to 29 °C 24 h AEL. Exceptions were crosses of *ey^3.5-Gal80; fas3^NP1233-Gal4; UAS-Dcr2 UAS-cd8GFP* and *UAS-omb^IR* or *UAS-tkv^IR* animals, which were kept at 18 °C before shifting to 29 °C at the early 3rd instar larval stage. For *UAS-NRT-wg*[33] gain-of-function experiments, *wg{KO;Gal4}/CyO; UAS-FLP* flies were crossed with *wg{KO;FRT wg^+ FRT NRT-wg} UAS-NRT-wg; tubP-Gal80^ts* flies and kept at 18 °C. While experimental animals were shifted to 29 °C at the 1st instar larval stage, control animals were maintained at 18 °C until the mid 3rd larval stage and then shifted to 29 °C. *wg{KO;FRT wg^+ FRT NRT-wg}* flies are described in ref. [30]. For GPC-specific *wg{KO;FRT NRT-wg FRT wg^+}* allele switching rescue experiments[30], *wg{KO;FRT NRT-wg FRT wg^+}/GlaBc; UAS-FLP* flies were crossed with (1) *wg{KO;NRT-wg}/GlaBc*[30] and (2) *wg{KO;NRT-wg}/GlaBc; R46E01-Gal4* and maintained at 25 °C. For gain-of-function experiments, *UAS-cd8-GFP; h^1J3-Gal4* was crossed with *UAS-arm^S10; dpp-lacZ^Exel.2* (ref. [62]), *UAS-FB1.1B^260b; R9B10-Gal4* was crossed with *UAS-N^intra* (from L. Tsuda, NCGG, Obu), and (1) *UAS-FB1.1B^260b; R9B10-Gal4* or (2) *wg{KO;NRT-wg}/GlaBc; R9B10-Gal4 UAS-FB1.1B^49b* were crossed with *UAS-omb* (♯2-1, ref. [63]; from G. Pflugfelder) and *wg{KO;NRT-wg}/GlaBc; UAS-omb*.

**Immunolabeling and imaging**. Brains were dissected in phosphate-buffered saline (PBS), fixed for 1 h at 20–24 °C in 2% paraformaldehyde (wt/vol) in 0.05 M sodium phosphate buffer (pH 7.4) containing 0.1 M L-lysine (Sigma-Aldrich) and washed in PBS containing 0.5% Triton X-100 (Sigma-Aldrich). Primary and secondary antibodies were diluted in 10% Normal Goat Serum (NGS) and PBT. The following primary antibodies were used: rabbit antibody to Ase (1:5000, from Y. N. Jan, HHMI, San Francisco[64]), rabbit antibody to Ato (1:5000, from Y. N. Jan[65]), mouse antibody to Brp (nc82, 1:10, Developmental Studies Hybridoma Bank [DSHB]), mouse antibody to Connectin (C1.427, 1:40, DSHB as marker for lobula plate layers 3/4[66]), guinea pig antibody to D (1:200, from A. Gould[67]), mouse antibody to Dac (mAbdac2-3, 1:50, DSHB), rabbit antibody to Dcp1 (#9578, 1:200, Cell Signaling Technologies), mouse antibody to Dlg (4F3, 1:50, DSHB), guinea pig antibody to Dpn (1:500, from J. Skeath, Washington University, St. Louis[68]), rat antibody to E-cad (DCAD2, 1:2, DSHB), mouse antibody to Fas3 (7G10, 1:5, DSHB), chicken, mouse, and rabbit antibodies to β-galactosidase (#ab9361, 1:500, Abcam; #Z3783, 1:300, Promega; #559762, 1:12,000, Cappel), rabbit antibody to GFP (#A6455, 1:1000, Molecular Probes), rabbit antibody to Hth (1:100, from R. Mann, Columbia University, New York[69]), rabbit antibody to Omb (1:400, G. Pflugfelder, University of Mainz/J.P. Vincent[63]), rabbit antibody to aPKC ς (sc-216, 1:100, Santa Cruz Biotechnologies), rabbit antibody to pSmad3 (pS423/425 #1880-S, 1:2, Epitomics), rabbit antibody to Tll (812, 1:20, J. Reinitz Segmentation Antibodies[70]), guinea pig antibody to Toy (1.170, 1:200, from U. Walldorf, University of Homburg), and mouse antibody to Wg (4D4, 1:20, DSHB). For immunofluorescence labeling, samples were incubated for 2.5 h at 20–24 °C in goat F(ab')₂ fragments coupled to FITC/DyLight 488, Cy3, or Alexa Fluor 647 (1:400; Jackson ImmunoResearch Laboratories): antibody to guinea pig (Cy3: #106-166-003; Alexa Fluor 647: #106-606-003), antibody to mouse (DyLight488: #115-486-003; Cy3: #115-166-003; Alexa Fluor 647: #115-606-003), antibody to rabbit (FITC: #111-096-003; Cy3: #111-166-003; Alexa Fluor 647: #111-606-003), antibody to rat (Cy3: #112-166-003). Furthermore, goat antibody to chicken IgY (H+L) (Alexa Fluor 555, Molecular Probes, #A21437, 1:400) was used. Images were collected with a Leica TCS SP5 II laser scanning confocal microscope and processed using Adobe Photoshop and Fiji software programs.

**Quantifications and statistics**. Statistical details of all experiments are reported in the figures and figure legends. To quantify T4/T5 neuron numbers, adult optic lobes were imaged in horizontal orientations and cell numbers were collected from three serial optical sections (6-μm distance) in five samples ($n = 15$) at the center of the optic lobe. Sample numbers and genotypes for all experiments are provided in Supplementary Tables 1 and 2. If not otherwise indicated, the penetrance of observed phenotypes was 100% for examined samples. Sample sizes were not predetermined by statistical calculations, but were based on the standard of the field. In a pool of control or experimental animals, specimens of the correct stage and genotype were selected randomly and independently from different vials. Data acquisition and analysis were not performed blinded but relied on samples with identified genotypes that were not limited in repeatability. The calculations of 95% confidence interval error bars and unpaired two-tailed Student's *t*-test *P* values were performed using Microsoft Excel software [Confidence.T and T.Test (type 3, not assuming equal variance)]. Prism 7 GraphPad was used to perform Shapiro-Wilk and D'Agostino-Pearson omnibus normality tests and data met the

assumption of normality in one or both tests. Quantifications are presented as scatter plots and bar graphs with means ±95% confidence interval error bars. $*P < 0.05$; $**P < 0.01$; $***P < 0.001$, $****P < 0.0001$.

**Data availability**. Image data sets generated and analyzed in this study are available from the corresponding author upon reasonable request. A source data file for quantifications shown in Figs. 1h and 8h is provided with this manuscript (Supplementary Data 1).

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

## Acknowledgements

We are grateful to C. Alexandre, A. Baena-Lopez, and J. P. Vincent (The Francis Crick Institute, London) for all Wg reagents and their invaluable advice for this study. We thank A. Brand (The Gurdon Institute, Cambridge), A. Gould (The Francis Crick Institute), Y. N. Jan (HHMI, San Francisco), R. Mann (Columbia University, New York), G. Pflugfelder (University of Mainz), F. Pignoni (SUNY Upstate Medical University), S. Sigrist (FU Berlin), J. Skeath (Washington University, St. Louis), L. Tsuda (NCGG, Obu), U. Walldorf (University of Homburg), the Bloomington *Drosophila* Stock Center, the *Drosophila* Genomics Resource Center, the Vienna *Drosophila* Resource Center, and the Developmental Studies Hybridoma Bank for fly strains and antibodies. We thank A. Yuen and W. Wang for help with Flybow and MARCM experiments. We thank H. Pynor for advice on the 3D drawing of the larval optic lobe. We thank J. P. Vincent and C. Alexandre, as well as A. Avola, C. de Miguel Vijandi, E. L. Powell, and R. Kaschula (The Francis Crick Institute, London) for critical reading of the manuscript. This work was supported by the Francis Crick Institute which receives its core funding from Cancer Research UK (FC001151), the UK Medical Research Council (FC001151) and the Wellcome Trust (FC001151), and by the UK Medical Research Council (U117581332).

## Author contributions

H.A. and I.S. conceived and designed the study. H.A. performed the experiments and analyzed the data. H.A. and I.S. prepared the manuscript.

## Additional information

**Competing interests:** The authors declare no competing interests.

