## [Peer Review File · Nature Communications]

Reviewers' comments:

Reviewer #1 (Remarks to the Author):

The manuscript by Apitz and Salecker describes how the first direction-selective cells in the fly visual system, T4 and T5, are specified during development. The study links early events between signaling centers to the production of layer specific T4 and/or T5 neurons.

Overall, the story presented is impressive, the manuscript is well-written and keeps an high experimental standard throughout the paper. The claims made in the paper are well supported by data. Furthermore, figures are very clear and schematics are helpful. In summary, this is a beautiful story that I strongly recommend for publication in Nature Communication. I only have the following minor comments:

“Motion-direction sensitive” is a strange way of saying “direction selective” and could cause confusion between motion sensitivity and direction selectivity. The standard term “direction selective” should be used.

The citation of literature describing motion – detecting circuits is a bit arbitrary and sometimes not precise. For example:

- Distinct sets of neurons in the lamina and medulla relay ON and OFF information to T4 and T5 neurons.: Why Maisak et al. and Yang et al.? Either cite reviews, or Joesch, Clark, Silies (for lamina neurons), Behnia, Fisher, Strother, Serbe (for medulla neurons)?
- The statement for preferred direction enhancement and null direction suppression is lacking the Leong et al. (2016, J Neurosc) citation, which was the first to show ND suppression. I would cite Haag 2016 instead of 2017, the Takemura EM studies don't show any enhancement or suppression...
- The orientation of dendritic arbors is shown in Takemura, 2013 and 2017, but not in Shinomiya et al. or Haag et al.

The first half of page 5 lacks references.

p. 7: wg expression in the s-IPC of $wg\{KO;NRT-wg\}$ flies was lost ...

This implies that it has been there at some point. Please clarify.

The Flybow approach in Figure 4d uses a different driver line, which also labels layers 1/2. Could you explain how this is an independent confirmation that layers 3/4 originate from the Dpp-expression domain?

Parts of the dac and ato paragraph (description of Figure 5 and Supp Fig 5) are misleading. The paragraph headline suggests that dac and ato mediate a specific switch to layer 1 and layer 2 innervating T4/T5 neurons. The phenotype associated with loss of dac and ato is a non-layer specific loss of T4/T5 neurons. Phenotypes of individual knockdowns lead to one Connectin-positive layer, which also is not showing a role of dac in “distinguishing layer 1/2 and 3/4 neuron subtypes” as suggested in the introduction to the experiment, and also cannot be clearly interpreted as “roles in distinction of layer 3 from layer 4 neurons”, which would need layer 3 / 4 specific markers. The concluding sentence that “downregulation of

Dac is required for specifying layer 3/4 innervating T4/T5 neurons" is a speculation and not a summary of the data shown.

These data are also overinterpreted in the discussion. If I am wrong, the authors should show / point out more clearly how Dac is specifically controlling layer 1/2 identity.

The order of the description of Notch-associated phenotypes is a bit confusing. The authors for example say that "T4 neurons were not impaired" in R17B05>>Su(H)IR, to then (after talking about the adult phenotype) say that layers 3/4 cannot be discriminated, which one could consider "impaired". It would be helpful to be more precise, and keep the full description of one phenotype together. Furthermore, could the authors explain how one knows what the oldest neuron are?

The discussion contains conclusions about layer-specificity and motion-directionality. It is impossible to conclude how the lobula plate phenotypes would affect direction selectivity (although this would be an interesting question) and should be discussed accordingly.

Reviewer #2 (Remarks to the Author):

The manuscript by Apitz and Salecker, "Spatio-temporal relays control layer specificity of motion-direction sensitive neurons in *Drosophila*", describes a series of signaling steps (via the secreted proteins wg and dpp and two transcription factors, omb and dac) leading to the distinct fates of two groups of related neurons in the developing optic lobe (T4/T5 neuron subtypes that project to different layers of the lobula plate and different types of distal cells such as C2 and C3). The authors previously showed that progenitor cells from a neuroepithelium called the p-IPC migrate to a different region (the d-IPC) where they form neuroblasts that produce T4/T5 neurons and distal cells in two time windows. They now identify molecular steps (initiated by diffusible wg from a different neuroepithelium) that specify an omb positive subregion of the p-IPC and identify subsets of T4/T5 neurons and distal cells that derive from precursors from this domain (with the remaining cell types in this group coming from other parts of the p-IPC). Some of the interest of this work comes from the specific neurons under study: T4/T5 cells have a well-documented critical role in fly motion vision and are an intriguing example of related neuronal cell types (four T4 and four T5 subtypes) with distinct anatomical features (e.g. layer patterns, dendrite orientation) and correlated functional differences - but very little is so far known about their development.

This is a carefully done study with high quality, very nicely displayed data. The results provide a new example of how the development of neuronal diversity in the optic lobes involves interactions of different precursor regions. As the first study that addresses aspects of T4/T5 subtype specification, this work is also relevant for thinking about how fly motion vision circuits develop and might have evolved.

Comments and suggestions:

1) The authors conclude that "Omb-mediated repression of Dac converts layer 1/2 into layer 3/4 T4/T5 neurons". Dac is shown to be downregulated in the T4/T5 neurons that project to layers 3 and 4 and this repression is mediated by omb. But what is the evidence that dac downregulation is actually necessary and/or sufficient to switch T4/T5 neurons from a layer 1/2 to a layer 3/4 fate? The reported dac loss-of function phenotypes appear to affect other aspects of T4/T5 differentiation and omb could in principle act via additional, currently unidentified targets.

2) When first reading the title ("Spatio-temporal relays control layer specificity of motion-direction sensitive neurons in Drosophila"), I thought this manuscript might be about neuronal interactions during layer formation (similar to e.g. previous work from the Salecker lab on medulla development) rather than the cell fate specification of neuronal cell types with different layer patterns. The link to layer specificity appears to be very indirect, for example, except for the final step (dac downregulation) the signaling relay that sets up the T4/T5 subtype differences also distinguishes C2 from C3 cells yet these don't have any arbors in lobula plate layers (but differ in other features). So perhaps emphasizing the specification of closely related neuronal subtypes (as is already done in some places in the manuscript) instead of layer specificity would be more accurate. (For example, an alternative title could perhaps be something like "Spatio-temporal relays control subtype specification of motion-direction sensitive neurons in Drosophila").

3) Related to the previous point: in the final paragraph of the introduction, the authors contrast potential mechanisms of T4/T5 subtype specification. It wasn't clear to me that the listed alternatives are mutually exclusive or even refer to comparable developmental steps, i.e. is this section about mechanism of layer formation (as references to guidance molecules seem to suggest) or cell fate specification? For example, the cited birth-order dependent growth cone segregation of R7 and R8 cells coexists with earlier mechanisms in the eye disc that determine R7 and R8 fate. Some rewording might help.

4) From the perspective of how the T4/T5 cell types may have evolved (see e.g.. Shinomyia et al 2015) it is interesting to see that the difference between the layer 12 and layer 34 T4/T5 cells is pre-specified in the p-IPC while T4 and T5 neuron appear to only become distinct later. The summary schematic in Figure 7 seems to imply that the differences between the remaining subtypes (layer 3 vs layer 4 cells, layer 1 vs layer 2 cells) are also determined at a later, post-p-IPC stage. Is this correct? Could the authors discuss more directly on when in the developmental sequence the remaining subtype differences emerge (or why conclusions about this are not yet possible, if this was the case)? The authors may already have the data to say more about this point or perhaps could carry out additional clonal analyses (e.g. using flybow or Twin-spot MARCM) to address this.

Other comments:

5) in the abstract: "Their axons innervate four lobula plate layers tuned to the four cardinal motion directions." For a reader not familiar with T4/T5 cells it may not be clear from this that different subtypes project to different layers.

6) p.3 "They represent the most numerous subtypes of the optic lobe (~5300 neurons/hemisphere)." Perhaps introduce the eight subtypes first? Cells of a single T4/T5 subtype (or type, this is not really a well-defined distinction) are not actually more numerous than many other optic lobe cell types.

7) p.8, last line of second paragraph "is essential" : is sufficient ?

8) There are some references to possible molecular determinants of layer 3 vs layer 4 T4/T5 fates but nothing is clearly shown. Genetic markers that distinguish between layer 3 and layer 4 T4 or T5 cells have been used in functional studies (Haag et al 2017, Fisher et al 2015) and could be used to try to clarify these observations. Also, did the authors check whether the difference between the layer 1 and layer 2 cells is also affected in these experiments? I am curious whether there are any indications as to whether the mechanisms that subdivide the layer 1/2 and layer 3/4 groups are the same or distinct.

9) Connectin is used as a marker for layers 3 and 4. It looks like a candidate for a cell surface protein that is expressed in T4/T5, regulated by omb/dac and that may contribute to layer development (i.e. this could be a link between cell fate and layer formation). Have the authors examined Con phenotypes?

10) Are the three and two LP layer phenotypes of the tethered-wg flies in Figure 1 "all or nothing" or are there also brains with incomplete layers (the counts appear not to exclude this possibility)? For example, some parts of the third layer in Fig 1f look as if they might locally consist of two layers and part of the layers may be missing. Similarly, in Fig 1g there appear to be perhaps a few patches of a third layer. The three layer phenotype is another case where it may be interesting to look at a marker that distinguishes layer 3 and layer 4 T4/T5 cells (see above).

Reviewer #3 (Remarks to the Author):

Defined neural circuits in the optic ganglia of the fruit fly allow the motion perception. Recent functional work defined the T4/T5 neurons as critical forming defined layers in the lobula plate that confer the 4 cardinal movement directions. In the current study Apitz and Salecker identify core aspects of the genetic program are involved in T4/T5 development. This thorough study starts out with the description of a membrane-tethered wingless knock-in resulting in an apparent loss of lobula plate layer 3/4. Next several experiments temporally restoring or removing wg (or tethered wg) and wg receptors provide support for requirement of Wg in this process. Wg appears required to induce dpp-expression in cells of the posterior inner proliferation centre (pIPC), that in turn appear to give rise to layer 3/4 T4/T5s (while interestingly layer 1/2 T4/T5s do not appear to emerge from the same set of

precursors). During subsequent developmental steps the interaction of Ato/Dac and Omb appear to be critical for proper differentiation of these cells, which similarly to most other neurons depends on asymmetric Notch activity (Notch on in T5 and Notch off in T4).

This is an excellent, very well organised manuscript, thus easily to follow when reading. Overall the data provided support the logic and interpretation of the authors.

Specific comments:

- Most specific wingless alleles used were published by Jean-Paul Vincent's lab a few years ago. They indeed provide an excellent genetic tool-set to study the role of wg in the context of the current manuscript (most of the alleles are nicely depicted with a cartoon). It would be beneficial to know the precise genotypes used, as in several instances several transgenes are used for the experiments. As it may not be easily done in the text a supplemental table could be used. The heterozygous wg-KO;Gal4 line could furthermore be used to assess wildtype wingless expression.

- The authors describe a roughly 25% or 50% reduction of T4/T5 neurons in wg knock-in. This virtually linear additive effect is not further discussed. Similar about half of the flies showed residual dpp-expression. It appears likely that at least part of the mechanism is not completely resolved, thus it would be beneficial to either experimentally address this issue or to provide explanations that fit the developmental model.

- While Wg appears to promote dpp in the precursors it remains not well explained how Dpp is involved in T4/T5 development. The only functional evidence is the knock-down of Tkv, which appear to have no effect on dpp-LacZ expression but results in loss of layer 3/4. The statement in the abstract that Dpp switches between layer 1/2 and layer 3/4 by regulating Omb is not sufficiently well documented. This may be resolved by providing additional support for dpp or to rephrase the corresponding sections.

Point by point response to comments of reviewers

We would like to thank the reviewers for their positive assessment of our work and hope to have addressed their very insightful comments satisfactorily to strengthen our manuscript.

Reviewer #1 (Remarks to the Author):

The manuscript by Apitz and Salecker describes how the first direction-selective cells in the fly visual system, T4 and T5, are specified during development. The study links early events between signaling centers to the production of layer specific T4 and/or T5 neurons.

Overall, the story presented is impressive, the manuscript is well-written and keeps an high experimental standard throughout the paper. The claims made in the paper are well supported by data. Furthermore, figures are very clear and schematics are helpful. In summary, this is a beautiful story that I strongly recommend for publication in Nature Communication. I only have the following minor comments:

“Motion-direction sensitive” is a strange way of saying “direction selective” and could cause confusion between motion sensitivity and direction selectivity. The standard term “direction selective” should be used.

We fully agree with this comment and have changed “sensitive” to “selective” in the title and text of our manuscript.

The citation of literature describing motion – detecting circuits is a bit arbitrary and sometimes not precise. For example:

- Distinct sets of neurons in the lamina and medulla relay ON and OFF information to T4 and T5 neurons.: Why Maisak et al. and Yang et al.? Either cite reviews, or Joesch, Clark, Silies (for lamina neurons), Behnia, Fisher, Strother, Serbe (for medulla neurons)?
- The statement for preferred direction enhancement and null direction suppression is lacking the Leong et al. (2016, J Neurosc) citation, which was the first to show ND suppression. I would cite Haag 2016 instead of 2017, the Takemura EM studies don't show any enhancement or suppression...
- The orientation of dendritic arbors is shown in Takemura, 2013 and 2017, but not in Shinomiya et al. or Haag et al.

We are grateful for the advice to use more appropriate citations in the introduction of our manuscript. We changed the references as summarized below:

- (1) “Distinct neuron sets in the lamina and medulla relay ON and OFF information to T4 and T5 neurons^{2,13}.” We replaced Maisak et al. (2013) and Yang et al. (2016) with the review article of Silies et al. (2014) to complement the Mauss et al. (2017) review.
- (2) “Direction-selectivity emerges within T4/T5 dendrites and involves the non-linear integration of input from these upstream neurons for enhancement in the preferred direction and suppression in the null-direction.”: We included Leong et al. (2016), replaced Haag et al. (2017) by Haag et al (2016) and removed Takemura et al. (2017), while keeping Fisher et al. (2015) and Strother et al. (2017).
- (3) “Dendritic arbors of the four T4 neuron subtypes have characteristic orientations that correlate with the direction preference of lobula plate layers innervated by their axons.”: We kept Takemura et al. (2013) and (2017), while removing Shinomiya et al. (2014) and Haag et al. (2016).

The first half of page 5 lacks references.

We had added solely two citations to the first half of page 5 as it summarized the findings from our two previous manuscripts, and we tried not to repeat these. As this was clearly too limited, we added a reference for neuron subtypes generated by the IPC (Hofbauer and Campos-Ortega, 1990) and for C & T neuron subtypes (Fischbach and Dittrich, 1989), as well one additional reference for our 2015 study in the revised manuscript.

p. 7: wg expression in the s-IPC of wg{KO;NRT-wg} flies was lost ...
This implies that it has been there at some point. Please clarify.

As *Wg* expression is indeed not induced, we have changed the wording to: “Furthermore, *wg* expression in the s-IPC of *wg{KO;NRT-wg}* mid 3rd instar larvae was absent when *wg* normally is detected (Fig. 2e,f).”

The Flybow approach in Figure 4d uses a different driver line, which also labels layers 1/2. Could you explain how this is an independent confirmation that layers 3/4 originate from the *Dpp*-expression domain?

Our aim was to use the Flybow approach as an independent confirmation for the differential origin of neurons innervating layer 1/2 and 3/4. To clarify the meaning of the statement in our manuscript, we adjusted the wording as follows: “Finally, we used the Flybow approach to label lineages in different colors. All clones contained T4/T5 neurons innervating either layers 1/2 or 3/4 but not both (Fig. 4d; *n*=43 clones in 26 optic lobes), consistent with distinct origins of these neuron subtypes.”

Parts of the *dac* and *ato* paragraph (description of Figure 5 and Supp Fig 5) are misleading. The paragraph headline suggests that *dac* and *ato* mediate a specific switch to layer 1 and layer 2 innervating T4/T5 neurons. The phenotype associated with loss of *dac* and *ato* is a non-layer specific loss of T4/T5 neurons. Phenotypes of individual knockdowns lead to one Connectin-positive layer, which also is not showing a role of *dac* in “distinguishing layer 1/2 and 3/4 neuron subtypes” as suggested in the introduction to the experiment, and also cannot be clearly interpreted as “roles in distinction of layer 3 from layer 4 neurons”, which would need layer 3 / 4 specific markers. The concluding sentence that “downregulation of *Dac* is required for specifying layer 3/4 innervating T4/T5 neurons” is a speculation and not a summary of the data shown. These data are also overinterpreted in the discussion. If I am wrong, the authors should show / point out more clearly how *Dac* is specifically controlling layer 1/2 identity.

Our findings indicate that *dac* and *ato* are required for the switch from T2/T3 to T4/T5 neuron formation irrespective of their layer-specificity. Consistently, *Dac* is initially expressed in all T4/T5 neurons. *Dac* is subsequently downregulated in layer 3/4 neurons and maintained in layer 1/2 neurons. This suggests that layer 1/2 identity could represent the default neuron fate. In support of this model, we present data in subsequent sections, demonstrating that *Omb* is necessary and sufficient to downregulate *Dac* and thus to generate T4/T5 neurons with layer 3/4 identity. We entirely agree that our wording requires adjustment to the data shown in this paragraph and Fig. 5 and Supp. Fig. 5. We also agree that only the use of specific markers distinguishing layer 3 and 4 T4/T5 neuron subtypes would enable us to correctly interpret phenotypes, in which only one Connectin-positive layer is visible in the lobula plate. Therefore, we carefully reworded the text as follows:

- On page 11, we changed the paragraph headline to: “*dac* and *ato* mediate the transition to T4/T5 neuron formation”.
- On page 11, we changed the sentence “While *dac* or *ato* single knockdown revealed potential roles in the distinction of layer 3 from layer 4 neurons (Supplementary Fig. 5h–j), simultaneous knockdown caused striking defects.” to “In samples with IPC-specific single *dac* or *ato* knockdown, frequently only one Connectin-positive lobula plate layer was discernible (Supplementary Fig. 5h–j). However, simultaneous *dac* and *ato* knockdown caused the loss of neurons with T4/T5 neuron morphologies and in consequence an undersized lobula plate neuropil.”
- On page 12, we changed the concluding paragraph to: “Therefore, *dac* and *ato* are required for the transition between Nb competence states in the d-IPC and the switch from T2/T3 to T4/T5 neuron formation. *Dac* is maintained in layer 1/2 innervating T4/T5 neurons and downregulated in layer 3/4 innervating T4/T5 neurons, suggesting that layer 1/2 identity could represent the default state.”
- We adjusted the text in the discussion on page 17 to: “*Dac* is initially expressed in all T4/T5 neurons but only maintained in layer 1/2 innervating subtypes. This suggests that an essential step for the specification of layer 3/4 innervating neurons is the downregulation of *Dac* and the suppression of the T4/T5 default neuron fate, i.e. layer 1/2 identity.”
- We changed the title of Figure 5 to: “*dac* and *ato* are required for T4/T5 neuron formation”.

As reviewer #2 had a related question to the requirement of *dac*, please also see our reply below.

The order of the description of Notch-associated phenotypes is a bit confusing. The authors for example say that “T4 neurons were not impaired” in *R17B05>>Su(H)IR*, to then (after talking about

the adult phenotype) say that layers 3/4 cannot be discriminated, which one could consider “impaired”. It would be helpful to be more precise, and keep the full description of one phenotype together. Furthermore, could the authors explain how one knows what the oldest neuron are?

- As suggested, we rearranged this paragraph to better group the description of phenotypes. Moreover, we replaced “T4 neurons were not impaired” with “T4 neurons were present”.
- The anterior medulla region is innervated by early born (the oldest) medulla neurons and T4 neurons. To clarify our sentence, we replaced “the oldest T4 neurons” by “T4 neurons connecting to the anterior proximal medulla”.

The discussion contains conclusions about layer-specificity and motion-directionality. It is impossible to conclude how the lobula plate phenotypes would affect direction selectivity (although this would be an interesting question) and should be discussed accordingly.

We agree that it is not possible to make any conclusions about behavioral outcomes. Therefore, we included the following sentence on page 18 of the discussion: “Finally, future behavioral studies of layer 3/4-deficient flies will address to what extent motion-direction selectivity is affected or compensatory mechanisms are in place.”

Reviewer #2 (Remarks to the Author):

The manuscript by Apitz and Salecker, “Spatio-temporal relays control layer specificity of motion-direction sensitive neurons in *Drosophila*”, describes a series of signaling steps (via the secreted proteins *wg* and *dpp* and two transcription factors, *omb* and *dac*) leading to the distinct fates of two groups of related neurons in the developing optic lobe (T4/T5 neuron subtypes that project to different layers of the lobula plate and different types of distal cells such as C2 and C3). The authors previously showed that progenitor cells from a neuroepithelium called the p-IPC migrate to a different region (the d-IPC) where they form neuroblasts that produce T4/T5 neurons and distal cells in two time windows. They now identify molecular steps (initiated by diffusible *wg* from a different neuroepithelium) that specify an *omb* positive subregion of the p-IPC and identify subsets of T4/T5 neurons and distal cells that derive from precursors from this domain (with the remaining cell types in this group coming from other parts of the p-IPC). Some of the interest of this work comes from the specific neurons under study: T4/T5 cells have a well-documented critical role in fly motion vision and are an intriguing example of related neuronal cell types (four T4 and four T5 subtypes) with distinct anatomical features (e.g. layer patterns, dendrite orientation) and correlated functional differences - but very little is so far known about their development.

This is a carefully done study with high quality, very nicely displayed data. The results provide a new example of how the development of neuronal diversity in the optic lobes involves interactions of different precursor regions. As the first study that addresses aspects of T4/T5 subtype specification, this work is also relevant for thinking about how fly motion vision circuits develop and might have evolved.

Comments and suggestions:

1) The authors conclude that “*Omb*-mediated repression of *Dac* converts layer 1/2 into layer 3/4 T4/T5 neurons”. *Dac* is shown to be downregulated in the T4/T5 neurons that project to layers 3 and 4 and this repression is mediated by *omb*. But what is the evidence that *dac* downregulation is actually necessary and/or sufficient to switch T4/T5 neurons from a layer 1/2 to a layer 3/4 fate? The reported *dac* loss-of function phenotypes appear to affect other aspects of T4/T5 differentiation and *omb* could in principle act via additional, currently unidentified targets.

A related concern regarding the role of *dac* has been raised by Reviewer #1 and we adjusted the text describing our findings in Fig. 5 and Supplementary Fig. 5 (please see above). Our data are consistent with the notion that *dac* is required for the formation of T4/T5 neurons, which adopt the default layer identity 1/2. Following *omb* knockdown, *Dac* expression is maintained and all neurons show layer 1/2 identity. Currently, we cannot exclude an additional role of *omb* in the specification of layer 3/4 neurons. To clarify this point, we included the following sentence on page 19 of the discussion: “It will

also be important to determine whether layer 3/4 specification is mediated solely by *Dac* downregulation, or whether *omb* has additional instructive roles.”

To further elucidate the requirement of *dac* in T4/T5 neurons, we have tested several experimental strategies in addition to the presented approaches in our manuscript. First, to assess a direct requirement in T4/T5 neurons, we knocked down *dac* with the help of *R9B10-Gal4* and *R17B05-Gal4*. *R9B10-Gal4* caused strong phenotypes in *omb* knockdown experiments, consistent with a requirement of *omb* in postmitotic T4/T5 neurons (Figs. 7m; 8). However, it reduced *Dac* expression only partially during the third instar larval stage, likely because of the late onset and relative strength of this driver in combination with the *UAS-dac* RNAi line. We observed increased expression of *Connectin* in the medulla and lobula but no lobula plate layer defects in adults (see Reviewer Figure 1a–d). *R17B05-Gal4* led to a more complete *Dac* knockdown. However, this driver is active earlier, and resulting phenotypes were consistent with a disruption of T4/T5 neuron identity, similar to those observed with *fas3^{NP1233}-Gal4* (see Supplementary Fig. 5i). Our observations suggest that *Dac* levels are tightly regulated to ensure that neurons adopt T4/T5 identity prior to segregating into layer 1/2 and 3/4 identities. Second, we attempted to prevent *Dac* downregulation in layer 3/4 T4/T5 neurons by over-expressing *UAS-dac* (an established functional transgene) in all T4/T5 neurons using *R9B10-Gal4*. We observed that even ectopic *Dac* was downregulated by endogenous *Omb* in layer 3/4 neurons (see Reviewer Fig. 1). Therefore, despite testing several genetic approaches, it was not possible to find means mirroring the manipulations of *Omb* to alter *Dac* expression with similar precision. As these findings may be due to technical limitations, we would prefer to not include these panels into our manuscript. We added some of this information in the discussion on page 18: “Interestingly, we did not succeed in preventing *Dac* downregulation in layer 3/4 neurons by over-expressing *Dac* (H.A., I.S., unpublished observations)”

Reviewer Figure 1. Additional analysis of *dac* function. (a–d) Knockdown of *dac* with *R9B10-Gal4* and *UAS-dac^{IR} KK106040*. Compared to controls (a), *Dac* protein levels (red) are partially reduced in T4/T5 neurons following the knockdown of *dac* with above transgene combination (b) at the third instar larval stage. Compared to controls (c), lobula plate layer formation appears normal, but *Connectin* levels in medulla and lobula neuropil layers are increased following *dac* knockdown (d) in adults. (e,f) Over-expression of *dac* with *R9B10-Gal4* in T4/T5 neurons.

Similar to controls (e), Dac (red) is downregulated in T4/T5 neuron subsets (arrowheads) following *R9B10-Gal4* mediated *UAS-dac* transgene expression (f). Scale bars, 50 μ m.

2) When first reading the title (“Spatio-temporal relays control layer specificity of motion-direction sensitive neurons in *Drosophila*”), I thought this manuscript might be about neuronal interactions during layer formation (similar to e.g. previous work from the Salecker lab on medulla development) rather than the cell fate specification of neuronal cell types with different layer patterns. The link to layer specificity appears to be very indirect, for example, except for the final step (dac downregulation) the signaling relay that sets up the T4/T5 subtype differences also distinguishes C2 from C3 cells yet these don’t have any arbors in lobula plate layers (but differ in other features). So perhaps emphasizing the specification of closely related neuronal subtypes (as is already done in some places in the manuscript) instead of layer specificity would be more accurate. (For example, an alternative title could perhaps be something like “Spatio-temporal relays control subtype specification of motion-direction sensitive neurons in *Drosophila*”).

We agree that our manuscript elucidates the mechanisms controlling the specification of neuron subtypes. Layer specificity is a core feature of motion-direction selective neuron identity and was our main read-out to assess cell fate decisions. We therefore would like to keep the link to layers and changed the title to “Spatio-temporal relays control layer identity of motion-direction selective neuron subtypes in *Drosophila*”.

3) Related to the previous point: in the final paragraph of the introduction, the authors contrast potential mechanisms of T4/T5 subtype specification. It wasn’t clear to me that the listed alternatives are mutually exclusive or even refer to comparable developmental steps, i.e. is this section about mechanism of layer formation (as references to guidance molecules seem to suggest) or cell fate specification? For example, the cited birth-order dependent growth cone segregation of R7 and R8 cells coexists with earlier mechanisms in the eye disc that determine R7 and R8 fate. Some rewording might help.

We introduced the two mentioned mechanisms as primarily postmitotic strategies that could control layer-specific T4/T5 neuron subtype identity to contrast these with our findings that subtype specification is determined as early as in neuroepithelial cells of the p-IPC. We agree that the described postmitotic mechanisms are not mutually exclusive. To avoid potential misunderstandings we adjusted the paragraph on page 5 as follows: “T4/T5 neuron diversity resulting in differential layer-specificity could be achieved by postmitotic combinatorial transcription factor codes upstream of distinct guidance molecules. Although not mutually exclusive, layer-specificity of T4/T5 neurons could also be determined by temporal differences in the expression of common postmitotic determinants, similar to the birth-order dependent R-cell growth cone segregation strategy described in the medulla^{28,29}. Here, we provide evidence for another mechanism, whereby layer-specific T4/T5 neuron subtype identity is determined early in the p-IPC neuroepithelium.”

4) From the perspective of how the T4/T5 cell types may have evolved (see e.g.. Shinomyia et al 2015) it is interesting to see that the difference between the layer 12 and layer 34 T4/T5 cells is pre-specified in the p-IPC while T4 and T5 neuron appear to only become distinct later. The summary schematic in Figure 7 seems to imply that the differences between the remaining subtypes (layer 3 vs layer 4 cells, layer 1 vs layer 2 cells) are also determined at a later, post-p-IPC stage. Is this correct? Could the authors discuss more directly on when in the developmental sequence the remaining subtype differences emerge (or why conclusions about this are not yet possible, if this was the case)? The authors may already have the data to say more about this point or perhaps could carry out additional clonal analyses (e.g. using flybow or Twin-spot MARCM) to address this.

We performed a large set of T4/T5-specific Flybow experiments (see Fig. 4d) but did not observe any clonal patterns revealing how layers 1/2 and 3/4 are further subdivided. Our experimental analysis is currently limited by the availability of markers sufficiently specific for T4/T5 neuron subtypes innervating a single lobula plate layer with an early onset of expression (see also below). However, we are in the process of screening for these markers and aim to address this important question in the future. Since Dac, Ato, and Notch signaling are all active within the d-IPC it is possible that, for example, asymmetric Nb or GMC divisions contribute to this process. In support of this notion, we observed that manipulations of these determinants/pathways lead to phenotypes with solely one detectable Connectin-positive layer. To address this comment, we added the following sentence to the

discussion on page 19: “While we did not explore how layer 1 and 2 neurons or layer 3 and 4 neurons become distinct from each other because of the lack of specific markers, our data suggest a possible contribution of Ato/Dac and Notch signaling, as these are active within the d-IPC.”

Other comments:

5) in the abstract: “Their axons innervate four lobula plate layers tuned to the four cardinal motion directions.” For a reader not familiar with T4/T5 cells it may not be clear from this that different subtypes project to different layers.

We changed this sentence in the abstract to: “Their axons innervate one of four lobula plate layers.”

6) p.3 “They represent the most numerous subtypes of the optic lobe (~5300 neurons/hemisphere).” Perhaps introduce the eight subtypes first? Cells of a single T4/T5 subtype (or type, this is not really a well-defined distinction) are not actually more numerous than many other optic lobe cell types.

This is true, we were side-tracked by the abundance of T4/T5 neuron subtypes in the lobula plate. We changed this sentence to: “Each optic lobe hemisphere contains ~5300 T4/T5 neurons¹¹.”

7) p.8, last line of second paragraph “ is essential” : is sufficient ?

We would prefer to keep the term essential, because the allele switching strategy is technically a rescue experiment to determine a requirement in a cell-type or area. We believe to be “sufficient”, a gene would need to be expressed ectopically, where it is not normally expressed while rescuing a phenotype.

8) There are some references to possible molecular determinants of layer 3 vs layer 4 T4/T5 fates but nothing is clearly shown. Genetic markers that distinguish between layer 3 and layer 4 T4 or T5 cells have been used in functional studies (Haag et al 2017, Fisher et al 2015) and could be used to try to clarify these observations. Also, did the authors check whether the difference between the layer 1 and layer 2 cells is also affected in these experiments? I am curious whether there are any indications as to whether the mechanisms that subdivide the layer 1/2 and layer 3/4 groups are the same or distinct.

How T4/T5 neuron acquire layer 1 versus 2 or layer 3 versus 4 identities is an important next question. We have started to work on this topic, but believe that this would be a new project on its own right. We are currently screening for novel single-layer specific markers that are active throughout development, as the two suggested markers are only expressed in T4/T5 neurons after mid-pupal development and therefore not suitable for knockdown experiments. Furthermore, they are not as single layer-specific in our hands as described (see below). In parallel, we are searching for novel transcriptional regulators that could distinguish T4/T5 neurons and function upstream of cell surface/guidance molecules. Connectin labeling allowed us to assess separation of layers 3 and 4. However, due to the lack of an equivalent marker for layers 1 and 2, it is currently not possible to conclusively examine whether layers 1 and 2 are also affected in the samples, in which layers 3 and 4 are not subdivided.

9) Connectin is used as a marker for layers 3 and 4. It looks like a candidate for a cell surface protein that is expressed in T4/T5, regulated by omb/dac and that may contribute to layer development (i.e. this could be a link between cell fate and layer formation). Have the authors examined Con phenotypes?

We agree that Connectin would be a very good candidate cell surface molecule to control layer formation. Connectin could be a possible direct target of *omb* and might mediate layer-specific targeting of layer 3/4 neurons, because Connectin expression is absent following *omb* knockdown and ectopically expressed following *omb* over-expression. Using *connectin*^{Mi(Mic)M104393} (Reviewer Fig. 2a), we confirmed that Connectin is expressed in Omb-positive layer 3/4 T4/T5 neurons (Reviewer Fig. 2b). *connectin*^{IR 17898/GD} mediates strong knockdown in medulla neurons using the *ey-FLPout* approach (Reviewer Fig. 2c). However, we observed no obvious phenotype following T4/T5 neuron-specific *connectin* knockdown (Reviewer Fig. 2d,e). While Connectin expression in T4/T5 neuron dendrites appeared reduced in the medulla and lobula, we detected residual Connectin expression in lobula plate layers 3/4 (Reviewer Fig. 2f,g). This suggests that Connectin is expressed in additional neuron

subtypes innervating layers 3/4, such as lobula plate tangential cells or lobula plate intrinsic neurons, and may play a role in these cells. Due to the lack of cell type specific Gal4 lines that are active throughout development in these cells, we are currently unable to further pursue this observation, but aim to do this in the future, once suitable drivers have been found. As our data are so far inconclusive regarding Connectin function in T4/T5 neuron layer formation, we would prefer to not include these data in the current manuscript.

Reviewer Figure 2. Expression and functional analysis of *connectin*. (a) GFP (green) expression in *connectin*^{Mik(Mic)M104393} recapitulates Connectin protein expression. (b) GFP-positive (green, arrowheads) T4/T5 neurons in *connectin*^{Mik(Mic)M104393} are labeled with the layer 3/4 neuron-specific marker Omb (red). (c) *connectin*^{IR}^{17898/GD} mediates efficient Connectin (red) knockdown in GFP-labeled (green) medulla (Me) neurons using the *ey-FLPout* approach. Similar to controls (d), T4/T5 neurons innervate four lobula plate layers following *R9B10-Gal4 UAS-cd8GFP* mediated *connectin* knockdown (e). Compared to controls (f), Connectin levels (red) are reduced in T4/T5 neuron dendrites in the medulla and lobula (arrowheads), but residual protein expression is observed in lobula plate layers 3 and 4 in these flies (g). Scale bars, 50 μ m.

10) Are the three and two LP layer phenotypes of the tethered-wg flies in Figure 1 “all or nothing” or are there also brains with incomplete layers (the counts appear not to exclude this possibility)? For example, some parts of the third layer in Fig 1f look as if they might locally consist of two layers and part of the layers may be missing. Similarly, in Fig 1g there appear to be perhaps a few patches of a third layer. The three layer phenotype is another case where it may be interesting to look at a marker that distinguishes layer 3 and layer 4 T4/T5 cells (see above).

We fully agree with this insightful observation. In *wg*[*KO*; *NRT-wg*] flies with three-layered lobula plates, the remaining Connectin-positive layer shows sometimes gaps, and could consist of either layer 3 or 4 or a mixture of both. We detected residual *dpp-lacZ* expression in one or the other stream in half of the samples at the 3rd instar larval stage (see Suppl. Fig 3a). The gaps in Connectin labeling might be

caused by a disruption of the birth-order dependent retinotopic map formation of the lobula plate in the absence of a continuous supply of progenitor cells. However, we currently lack experimental evidence to support this idea. We found that the two suggested markers *VT050384-Gal4* (used in Haag et al. 2017) and *R54A03-Gal4* (used in Fisher et al. 2015) are not strictly layer 3-specific in our hands (Reviewer Fig. 3a,b). This may have become apparent because of strong *UAS-FB1.1B* reporter activity driving membrane-tethered GFP expression. In *wg[KO;NRT-wg]* flies carrying *R54A03-Gal4 UAS-FB1.1B* transgenes, we detected continuous labeling in layer 2 and patchy expression in the third remaining layer (Reviewer Fig. 3c), consistent with the notion that this layer contains a mixture of remaining T4/(T5) neuron subtypes normally innervating layers 3 and 4. However, since this marker shows also expression in layers 2 and 4 in wild type (Reviewer Fig. 3b), we are unable to make a clear statement about the subtype composition of the third layer in half of the *wg[KO;NRT-wg]* flies using this marker. To clarify this point we added the following sentence to the legend of Fig. 1: “Similar to *nc82* (f), Connectin labeling showed gaps in the remaining lobula plate layer (j), potentially consisting of both layer 3 and 4 neurons.”

Reviewer Figure 3. Layer-specific expression of *VT050384-Gal4* and *R54A03-Gal4*. *cd8GFP* expression (green) driven by *VT050384-Gal4 UAS-FB1.1B* (a) and *R54A03-Gal4 UAS-FB1.1B* (b) is enriched in lobula plate layer 3, but also detectable in layers 2 and 4. In *wg[KO;NRT-wg]* flies (c), *R54A03-Gal4 UAS-FB1.1B* driven *cd8GFP* expression (green) is detected in layer 2 and the third layer, consisting of layer 3 and possibly layer 4 neurons. Scale bars, 50 μ m.

Reviewer #3 (Remarks to the Author):

Defined neural circuits in the optic ganglia of the fruit fly allow the motion perception. Recent functional work defined the T4/T5 neurons as critical forming defined layers in the lobula pate that confer the 4 cardinal movement directions. In the current study Apitz and Salecker identify core aspects of the genetic program are involved in T4/T5 development. This thorough study starts out with the description of a membrane-tethered wingless knock-in resulting in an apparent loss of lobula plate layer 3/4. Next several experiments temporally restoring or removing *wg* (or tethered *wg*) and *wg* receptors provide support for requirement of *Wg* in this process. *Wg* appears required in induce *dpp*-expression in cells of the posterior inner proliferation centre (pIPC), that in turn appear to give rise to layer 3/4 T4/T5s (while interestingly layer 1/2 T4/T5s do not appear to emerge from the same set of precursors). During subsequent developmental steps the interaction of *Ato/Dac* and *Omb* appear to be critical for proper differentiation of these cells, which similarly to most other neurons depends on asymmetric Notch activity (Notch on in T5 and Notch off in T4).

This is an excellent, very well organised manuscript, thus easily to follow when reading. Overall the data provided support the logic and interpretation of the authors.

Specific comments:

- Most specific wingless alleles used were published by Jean-Paul Vincent's lab a few years ago. They indeed provide an excellent genetic tool-set to study the role of *wg* in the context of the current manuscript (most of the alleles are nicely depicted with a cartoon). It would be beneficial to know the precise genotypes used, as in several instances several transgenes are used for the experiments. As it may not be easily done in the text a supplemental table could be used. The heterozygous *wg*-*KO*;*Gal4* line could furthermore be used to assess wildtype wingless expression.

We agree with the reviewer that our study relies on a complex set of genetic tools and that it is essential to provide precise details about genotypes. We therefore had added two Supplementary Tables providing information about precise genotypes shown in each Figure panel in addition to sample numbers and penetrance of phenotypes for main and Supplementary Figures (please see the last pages of Supplementary Information). To make this more clear, we now added the following information on page 8 in the section employing the genetic tool-set of Jean-Paul Vincent's team: "(for full genotypes, see Supplementary Table 1)". Furthermore, we have used the heterozygous *wg*;*KO*;*Gal4* line to determine wild type *wg* expression throughout larval development in Fig. 2d,e and Supplementary Fig. 1.

- The authors describe a roughly 25% or 50% reduction of T4/T5 neurons in *wg* knock-in. This virtually linear additive effect is not further discussed. Similar about half of the flies showed residual *dpp*-expression. It appears likely that at least part of the mechanism is not completely resolved, thus it would be beneficial to either experimentally address this issue or to provide explanations that fit the developmental model.

The 25% and 50% reduction of T4/T5 neurons in adults can be explained by the observed partial penetrance of larval phenotypes. At this stage, half of the samples showed residual *dpp-lacZ* expression in one progenitor stream. These residual *dpp-lacZ* positive progenitor cells will generate roughly 25% less T4/T5 neurons since they only come from one stream instead of two in wild type. The 50% reduction of T4/T5 neurons in adults corresponds to samples with no *dpp-lacZ* expression at the larval stage. We added the following information to the discussion on page 15 to clarify this point: "Half of the *wg*;*KO*;*NRT-wg*] flies showed residual *dpp* expression in one progenitor stream at the 3rd instar larval stage and a 25% reduction of T4/T5 neurons, correlating with three lobula plate layers in adults. The other half lacked *dpp-lacZ* expression and showed a 50% reduction of T4/T5 neurons correlating with two remaining layers."

These observations however raise the important question about the possible causes for the observed variability in phenotypes. To address this issue, we have re-examined *NRT-wg* expression in *wg*;*KO*;*NRT-wg*] flies in a larger data set ($n=41$). Specifically, we asked whether in *wg*;*KO*;*NRT-wg*] flies, the *Wg*-positive neuroblast (Nb) clone adjacent to the dorsal p-IPC arm and the s-IPC adjacent to the ventral p-IPC arm are equally affected. The new results are very revealing: *NRT-wg* was absent in $n=22/41$ samples in both domains. However, a subset of samples showed weak residual expression in the s-IPC ($n=9/41$), the small Nb clone ($n=8/41$) or both ($n=2/41$). This matches the penetrance of loss or residual *dpp-lacZ* expression and of adult phenotypes in *wg*;*KO*;*NRT-wg*] flies. We added these novel data to Supplementary Fig. 2 and to the results on page 7: "However in the IPC (**Fig. 2b**; **Supplementary Fig. 2c-f**), *NRT-Wg* expression was absent in the s-IPC adjacent to the ventral p-IPC and the Nb clone adjacent to the dorsal p-IPC in approximately half of the samples ($n=22/41$). In the remaining samples, residual *NRT-Wg* expression was found either in the s-IPC ($n=9/41$), the Nb clone ($n=8/41$), or both ($n=2/41$)."

In the discussion of our originally submitted manuscript, we proposed that *NRT-wg* was likely able to partially substitute for *wg* because of the initial close proximity of these NE domains at the early 3rd instar larval stage. Consistently in our new data set, some samples showed residual *NRT-Wg* in the s-IPC or the Nb clone. However, not all cells expressed *NRT-Wg*. This suggests a cell-intrinsic threshold to activate *wg* expression that Wnt signaling has to reach. In samples with residual expression, *NRT-Wg* from the GPC areas was likely able to activate its own expression in single cells within the s-IPC by reaching this threshold. These individual cells however are not sufficient to induce an all-or-nothing

event in the s-IPC itself and to activate *NRT-wg* expression in all their neighbours as well. We added this explanation to the discussion on pages 16/17 as follows: “While this partial phenotypic penetrance is not fully understood, NRT-Wg likely partially substituted for Wg because of the initial close proximity of the GPC areas and the s-IPC and Nb clone. Occasional residual NRT-Wg expression in the s-IPC argues against an all-or-nothing inductive event and suggests a model, whereby cell-intrinsic signaling thresholds have to be reached.”

- While Wg appears to promote *dpp* in the precursors it remains not well explained how Dpp is involved in T4/T5 development. The only functional evidence is the knock-down of *Tkv*, which appear to have no effect on *dpp-LacZ* expression but results in loss of layer 3/4.

We realize that findings explaining how Dpp signaling controls T4/T5 development are not presented in one section but distributed across several main and Supplementary Figures. Moreover, we had previously reported a role for Dpp signaling in the p-IPC (Apitz and Salecker, 2015) and relied on this information in this manuscript. We had shown that Dpp signaling is required for epithelial-mesenchymal transition (EMT) of migratory progenitors from neuroepithelial (NE) cells in the Dpp-expression domain of the p-IPC. In *thickveins (tkv)* loss-of-function clones, *dpp-lacZ* was still expressed because it lies upstream of the receptor and because of perdurance of β -Galactosidase. However importantly, target gene expression was impaired: i.e. the negative regulator *brinker* was upregulated and the positive regulator *omb* absent.

In the current manuscript, we provide several lines of evidence to support the conclusion that Dpp-signaling mediates not only EMT of progenitors, but also the specification of layer 3/4 T4/T5 neurons within NE cells of the p-IPC. First, following IPC-specific knockdown of *tkv*, *dpp-lacZ*-positive NE cells fail to undergo EMT and the corresponding progenitor streams are absent (Fig. 3k). This in turn results in the loss of T4/T5 neurons innervating layers 3/4 in adults (Fig. 3n), as layer 3/4 T4/T5 neurons are specifically derived from the Dpp-expression domain (Fig. 4). Second, p-Mad labeling shows that Dpp-signaling is only active within NE cells in the Dpp-expression domain of the p-IPC (Fig. 7a), but not in the d-IPC or in T4/T5 neurons. Consistently, *tkv* knockdown within p-IPC NE cells results in the absence of layers 3/4 (Fig. 3n), while *tkv* knockdown in all d-IPC Nb and progeny, or in primarily postmitotic T4/T5 neurons has no effect (Supplementary Fig. 7d-f). Third, to reveal how Dpp signaling mediates the specification of layer 3/4 neurons within p-IPC NE cells we focused on *omb*, since we knew from our previous studies (and now provide independent confirmation, see below) that it is a Dpp-target gene in the p-IPC. In contrast to *tkv*, *omb* knockdown does not affect EMT in the Dpp-expression domain (Fig. 7h,i). Connectin-positive layers 3/4 are absent not only following *omb* knockdown in p-IPC NE cells (Fig. 7k), but also in the d-IPC and/or T4/T5 neurons (Fig. 7l,m). Therefore, *omb* mediates the Dpp-signaling dependent specification of layer 3/4 T4/T5 neurons.

The statement in the abstract that Dpp switches between layer 1/2 and layer 3/4 by regulating Omb is not sufficiently well documented. This may be resolved by providing additional support for *dpp* or to rephrase the corresponding sections.

Using *tkv* clonal analysis we had previously shown that *omb* is a Dpp-target gene in the p-IPC using *omb-lacZ* as a reporter (Apitz and Salecker, 2015). Reassessing our current manuscript in light of these comments, we realized that this information was not sufficiently emphasized and that we had not provided direct evidence for regulation of Omb protein by Dpp signaling. To further strengthen this conclusion, we included a new set of data in Fig. 7f,g, demonstrating that Omb expression in the p-IPC and progeny is absent following IPC-specific *tkv* knockdown, and added the following sentence to the results on page 13: “Consistently, Omb was also decreased following IPC-specific *tkv* knockdown (Fig. 7f,g).”

REVIEWERS' COMMENTS:

Reviewer #1 (Remarks to the Author):

The authors have responded to all my criticism. I would only still replace 'motion-direction selective' with 'direction selective' throughout the article - that is the commonly used term. I congratulate them on a beautiful study and fully support publication in Nature Communications.

Reviewer #2 (Remarks to the Author):

The authors have adequately addressed my earlier comments and suggestions. Congratulations on a very nice study !

Reviewer #3 (Remarks to the Author):

The authors have addressed or discussed all points that were raised by the reviewers. The manuscript has further improved in quality and provides an excellent study on the steps that coordinate the development of a motion circuit.

Point by point response to comments of reviewers

We would like to thank the reviewers for their very positive assessment of our study.

Reviewer #1 (Remarks to the Author):

The authors have responded to all my criticism. I would only still replace 'motion-direction selective' with 'direction selective' throughout the article - that is the commonly used term. I congratulate them on a beautiful study and fully support publication in Nature Communications.

As suggested, we replaced "motion-direction selective" by "direction-selective" throughout our manuscript including the title.

Reviewer #2 (Remarks to the Author):

The authors have adequately addressed my earlier comments and suggestions. Congratulations on a very nice study !

No response is required.

Reviewer #3 (Remarks to the Author):

The authors have addressed or discussed all points that were raised by the reviewers. The manuscript has further improved in quality and provides an excellent study on the steps that coordinate the development of a motion circuit.

No response is required.